# Debiased Model-based Representations for Sample-efficient Continuous Control

Jiafei Lyu [1]  Zichuan Lin [1]  Scott Fujimoto [2]  Kai Yang [1]  Yangkun Chen [1]  Saiyong Yang [1]  Zongqing Lu [3]
Deheng Ye [1]

## Abstract

Model-based representations recently stand out as a promising framework that embeds latent dynamics information into the representations for downstream off-policy actor-critic learning. It implicitly combines the advantages of both model-free and model-based approaches while avoiding the training costs associated with model-based methods. Nevertheless, existing model-based representation methods can fail to capture sufficient information about relevant variables and can overfit to early experiences in the replay buffer. These incur biases in representation and actor-critic learning, leading to inferior performance. To address this, we propose Debiased model-based Representations for Q-learning, tagged DR.Q algorithm. DR.Q explicitly maximizes the mutual information between the representations of the current state-action pair and the next state besides minimizing their deviations, and samples transitions with faded prioritized experience replay. We evaluate DR.Q on numerous continuous control benchmarks with a single set of hyperparameters, and the results demonstrate that DR.Q can match or surpass recent strong baselines, sometimes outperforming them by a large margin. Our code is available at https://github.com/dmksjfl/DR.Q.

## 1. Introduction

Reinforcement learning (RL) agents are known to suffer from sample inefficiency, often requiring large amounts of online interactions to learn a good policy, which can be expensive and hinder the practical application of RL algorithms. To improve the sample efficiency of RL agents,

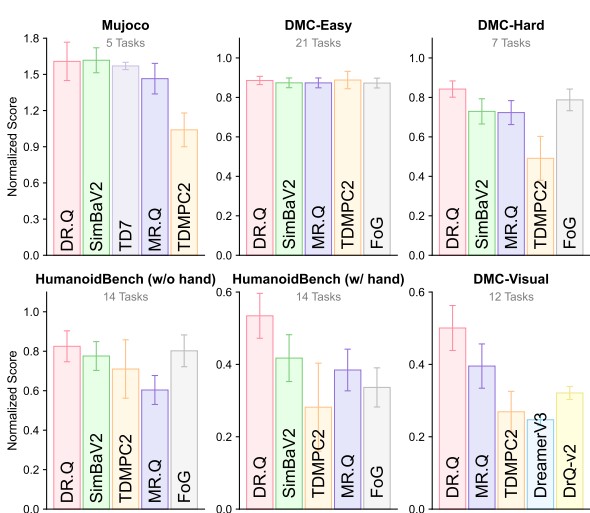

*Figure 1.* **Benchmark summary.** We aggregate results from three continuous control benchmarks and 73 tasks. Error bars denote the 95% confidence interval. DR.Q generally match or outperform strong baselines like SimBaV2, MR.Q, and TDMPC2.

previous works have explored numerous directions in model-free RL, such as mitigating the value overestimation issue (Fujimoto et al., 2018; Kuznetsov et al., 2020; Lyu et al., 2022), reusing data from the replay buffer (Chen et al., 2021; D'Oro et al., 2023; Lyu et al., 2024), modifying network architecture (Nauman et al., 2024; Lee et al., 2025a;b), etc. Meanwhile, some researchers resort to learning the world model of the environment and leveraging it for planning (Hansen et al., 2022; 2024) or data augmentation (Janner et al., 2019; Voelcker et al., 2025). These model-based methods can exhibit higher sample efficiency compared to model-free ones, but their training costs are often higher.

To incorporate the benefits of model-based objectives into model-free algorithms, recent works (Fujimoto et al., 2023; 2025) propose to learn *model-based representations* that train state and action representations by modeling the latent dynamics of the environment. The learned model-based representations are then fed forward to downstream actor and critic networks to learn the policy and value functions. This framework is promising since it enables richer alternative learning signals and faster adaptation to environmental

[1]Tencent Hunyuan [2]McGill University [3]School of Computer Science, Peking University. Correspondence to: Jiafei Lyu <dmksjfl@gmail.com>, Deheng Ye <775050607@qq.com>.

*Proceedings of the $43^{rd}$ International Conference on Machine Learning*, Seoul, South Korea. PMLR 306, 2026. Copyright 2026 by the author(s).

dynamics. However, we argue that there are two factors that negatively affect model-based representations. First, existing methods often train model-based representations by minimizing the deviation between the current state-action representation and the next state representation, which unfortunately does not necessarily incur higher mutual information between them (see Theorem 4.1). It indicates that current representation learning objectives may fail to capture sufficient information about the state-action representation and the next state representation. Second, the model-based representations are trained either by uniform sampling or prioritized experience replay (PER) which favors transitions with large temporal difference (TD) errors. Nevertheless, the learned representations can overfit to early (bad) experiences due to the primacy bias (Nikishin et al., 2022). These factors cause bias in representation learning and eventually incur inferior performance.

As such, we propose Debiased model-based Representations for Q-learning in this work, dubbed DR.Q algorithm. It actively maximizes the mutual information between the representations of the current state-action pair and the next state, apart from the commonly adopted objective of minimizing the representation deviations. By doing so, the learned state-action representation and the next state representation not only become numerically close, but also encode more relevant information about each other. Furthermore, DR.Q introduces a faded prioritized experience replay approach that assigns higher priority to new experiences with large TD errors and lower priority to earlier experiences. This generally ensures that more valuable samples are used for training while alleviating the influence of the primacy bias. Altogether, these result in informative model-based representations that can better benefit actor-critic learning.

We evaluate DR.Q across 73 environments from three standard continuous control online RL benchmarks: MuJoCo (Todorov et al., 2012), DMC suite (Tassa et al., 2018), and HumanoidBench (Sferrazza et al., 2024). These tasks feature diverse characteristics and varying complexities. As depicted in Figure 1, DR.Q can match or outperform strong domain-specific algorithms and general baselines, sometimes by a large margin. We open-source our code, model weights, and logs to facilitate future research.

## 2. Related Work

**Dynamics-based representation learning.** Representation learning (Bengio et al., 2013; Lesort et al., 2018) is an effective way to capture the underlying patterns of the data, where the intermediate features can be learned independently from downstream tasks. Many representation learning methods are used in RL to produce high-quality representations, e.g., contrastive representation learning methods (Laskin et al., 2020; Stooke et al., 2021; Zheng et al., 2023) and self-

supervised representation learning methods (Grill et al., 2020; Paster et al., 2021; Bardes et al., 2024; Garrido et al., 2024). Meanwhile, representation learning in RL is often related to dynamics, i.e., modeling how the system evolves from the current state given one legal action. Naturally, such dynamics-based representation learning that learns latent dynamics models can be found in numerous model-based RL papers (Watter et al., 2015; Finn et al., 2016; Zhang et al., 2019; Schrittwieser et al., 2020; 2021; Hansen et al., 2022; 2024; Karl et al., 2016; Hafner et al., 2019; 2023; Wang et al., 2024; Sun et al., 2024; Krinner et al., 2025). Moreover, dynamics-based representation learning is also explored in model-free methods, which learn representations by predicting future latent states (Munk et al., 2016; Van Hoof et al., 2016; Zhang et al., 2018; Gelada et al., 2019; Schwarzer et al., 2020; Lee et al., 2020; Ota et al., 2020; Guo et al., 2020; McInroe et al., 2021; Guo et al., 2022; Cetin et al., 2022; Yu et al., 2022; Zhao et al., 2023; Yan et al., 2024; Fujimoto et al., 2023; 2025; Ni et al., 2024; Scannell et al., 2024a;b; Bagatella et al., 2025). These works demonstrate the effectiveness and advantages of dynamics-based representation learning under various scenarios.

**Sample-efficient RL algorithms.** Sample efficiency is one of the key metrics for evaluating online RL agents. Higher sample efficiency is preferred since it means that agents can learn faster and better given a fixed budget of online interactions. Many efforts have been made to enhance sample efficiency, including improving the exploration ability of the agent (Still & Precup, 2012; Burda et al., 2018; Haarnoja et al., 2018; Ladosz et al., 2022; Yang et al., 2024; Jiang et al., 2025), scaling up compute by reusing data from the replay buffer (Chen et al., 2021; D'Oro et al., 2023; Lyu et al., 2024; Romeo et al., 2025), parallel simulation (Seo et al., 2025; Obando-Ceron et al., 2025), using normalization approaches (Wang et al., 2020; Gogianu et al., 2021; Lyle et al., 2024; Bhatt et al., 2024), mitigating value estimation bias (Van Hasselt et al., 2016; Fujimoto et al., 2018; Kuznetsov et al., 2020; Moskovitz et al., 2021; Lyu et al., 2022; 2023), leveraging model-based approaches (Janner et al., 2019; Buckman et al., 2018; Hafner et al., 2020; Lai et al., 2021; Fan & Ming, 2021; Wang et al., 2022; Voelcker et al., 2025; Wang et al., 2025c;a; Amigo et al., 2025), etc. Another line of study improves sample efficiency by modifying the network architecture and scaling network capacities (Nauman et al., 2024; Kang et al., 2025; Lee et al., 2025a;b; Lyu et al., 2026). Instead, DR.Q focuses on improving model-based representations without altering network configurations.

**Experience replay methods.** Off-policy RL methods often use uniform sampling during training (Mnih et al., 2015; Haarnoja et al., 2018; Fujimoto et al., 2018), i.e., all transitions in the replay buffer are treated equally. To better utilize the gathered samples, numerous experience replay methods have been developed. (Schaul et al., 2015) introduces pri-

oritized experience replay (PER), which assigns priority to transitions based on their TD errors. PER is shown to be effective and has inspired numerous subsequent works (Horgan et al., 2018; Fujimoto et al., 2020; Saglam et al., 2023; Pan et al., 2022; Oh et al., 2022; Li et al., 2024). Hindsight experience replay (HER) (Andrychowicz et al., 2017; Fang et al., 2019; Yang et al., 2021) mitigates the sparse reward issues by injecting additional goals into trajectories. Other valuable attempts include adjusting the sampling probability to make the sampling distribution more uniform (Yenicesu et al., 2024), organizing the experiences into a graph (Hong et al., 2022), and incorporating a "forget" mechanism that allocates lower probabilities to older experiences while sampling more recent experiences (Novati & Koumoutsakos, 2019; Wang et al., 2020; Kang et al., 2025), etc. The faded prioritized experience replay in DR.Q leverages the advantages of PER and the forget mechanism to ensure that more valuable samples are used for training.

## 3. Preliminary

**Reinforcement learning (RL).** RL problems can be formulated as a Markov Decision Process (MDP), which is specified by a 5-tuple $(\mathcal{S}, \mathcal{A}, r, \gamma, \mathcal{P})$, where $\mathcal{S}$ is the state space, $\mathcal{A}$ is the action space, $r$ is the reward, $\mathcal{P}$ is the dynamics function, $\gamma$ is the discount factor. RL agent aims to learn a policy $\pi : \mathcal{S} \to \mathcal{A}$ that maximizes the cumulative discounted return $\sum_{t=0}^{\infty} \gamma^t r_t$. RL algorithms learn a value function $Q^\pi(s, a) := \mathbb{E}[\sum_{t=0}^{\infty} \gamma^t r_t | s_0 = s, a_0 = a]$, which measures the expected return given state $s$ and action $a$.

**Model-based representations.** Model-based representations leverage objectives from model-based RL to learn implicit state-action (or state) representations by enforcing dynamics consistency in the latent space. To be specific, one needs to train the state encoder $f(\cdot)$, the state-action encoder $g(\cdot)$, and the reward function $\hat{r}(\cdot)$. The state encoder $f(\cdot)$ receives the state as the input and outputs the state representation, i.e., $z_s = f(s)$. The resulting state representation $z_s$ and the corresponding action $a$ are then fed into the state-action encoder $g(\cdot)$ and the reward function to output the state-action representation $z_{sa} = g(z_s, a)$ and the predicted reward $\hat{r}(z_s, a)$. Then, the model-based representations are trained by minimizing $\mathbb{E}[(z_{sa} - z_{s'})^2 + (\hat{r} - r)^2]$, where $z_{s'}$ is the next state representation. Typically, MRQ (Fujimoto et al., 2025) trains model-based representations by:

$$\mathcal{L}_{\text{enc}}^{\text{MR.Q}} = \sum_{i=1}^{H} \lambda_r \text{CE}(\hat{r}_i, \text{TwoHot}(r_i)) + \lambda_d \mathbb{E}[(\hat{z}_{s',i} - \tilde{z}_{s',i})^2]$$
$$+ \lambda_t \mathbb{E}[(\hat{d}_i - d_i)^2], \tag{1}$$

where CE is the cross entropy loss, $H$ is the encoder horizon, TowHot is the two-hot encoding, $\hat{d}$ is the predicted done flag, $d$ is the true done flag, $\lambda_r, \lambda_d, \lambda_t$ are coefficients that balance each loss term, the subscript $i$ denotes the corre-

sponding value at step $i$, $i \in [1, H]$. $\hat{z}_{s',i}$ is a linear mapping of the state-action representation $z_{sa,i}$, and $\tilde{z}_{s',i}$ is the next state representation produced by the target state encoder.

**Notations.** $H(X)$ denotes the entropy of the random variable $X$, and $H(X|Y)$ is the conditional entropy of $X$ given $Y$, $I(X;Y)$ is the mutual information between $X$ and $Y$.

## 4. Debiased Model-based Representations

In this section, we introduce our **D**ebiased model-based **R**epresentation learning for **Q**-learning, dubbed DR.Q algorithm. Following prior methods that learn model-based representations (Fujimoto et al., 2023; 2025), DR.Q separates the conventional actor-critic training process into two phases: (i) learning state representations $z_s$ and state-action representations $z_{sa}$ using model-based objectives, (ii) optimizing downstream value functions $Q_\theta$ parameterized by $\theta$ and the policy $\pi_\phi$ parameterized by $\phi$. To that end, DR.Q needs to train the following components:

$$f : s \to z_s; \quad g : (s, a) \to z_{sa}, \quad \hat{r} : z_{sa} \to \mathbb{R},$$
$$\pi_\phi : z_s \to a, \quad Q_\theta : z_{sa} \to \mathbb{R}. \tag{2}$$

The overall framework of DR.Q is presented in Figure 2.

### 4.1. Representation Learning with Mutual Information

In model-based RL, it is common practice to learn the dynamics model of the underlying environment, i.e., to train a model $g(s, a)$ that predicts the next state $s'$ given the current state $s$ and action $a$. The objective function gives $\min \mathbb{E}[(g(s, a) - s')^2]$, which fulfills the dynamics consistency in the raw state-action space. Following this, prior methods (Fujimoto et al., 2025; Hansen et al., 2024) often choose to minimize the deviation between the state-action representation $z_{sa}$ and the next state representation $z_{s'}$ when learning dynamics models in the latent space.

Such a training paradigm seems rational, but merely minimizing the numerical distance (e.g., Euclidean distance) between the representations $z_{sa}$ and $z_{s'}$ does not inherently provide a mechanism to discard redundant or irrelevant information. It is possible that the learned representations are only *falsely aligned*, where the small numerical distances are achieved by incorrectly minimizing the deviations between redundant elements, while key components that are vital for downstream value learning and policy learning may become less emphasized. This phenomenon can be pronounced for high dimensional state space or action space tasks, where many factors can be less important given a specific task (e.g., the dexterous hands information matters less for the humanoid robot to run or walk). We further theoretically justify our claim. For theoretical analysis, we assume that $S, A$ are random variables that follow some distribution (e.g., $S$ can follow a distribution over initial states and agent's actions). $S = s, A = a$ are the observed state and action

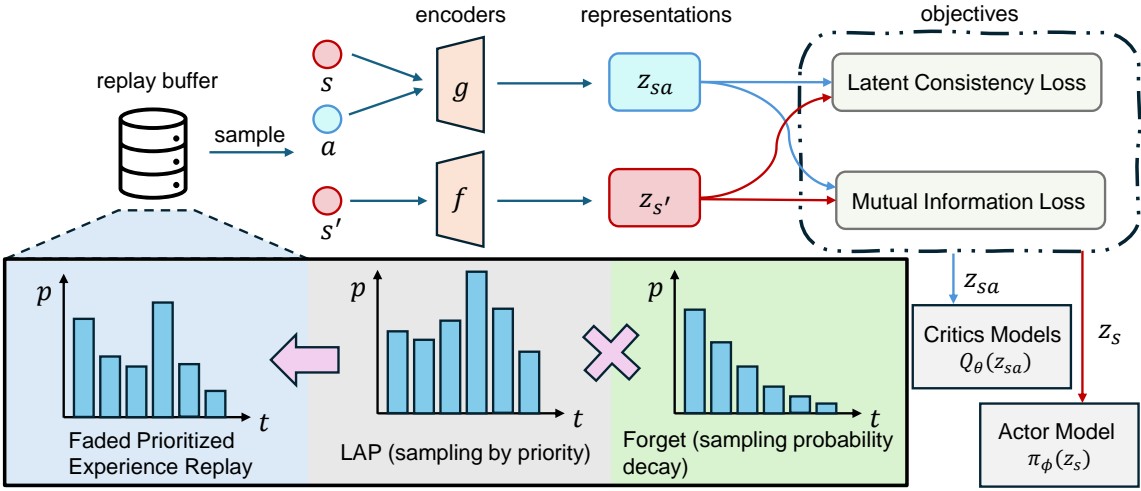

*Figure 2.* **Overall framework of DR.Q.** DR.Q introduces an auxiliary loss for maximizing the mutual information between the state-action representation $z_{sa}$ and the next state representation $z_{s'}$ rather than merely minimizing the latent dynamics consistency loss. Meanwhile, DR.Q improves the sampling strategy by combining the prioritized experience replay with the experience forget mechanism.

vectors. We denote $Z_{sa}$ and $Z_{s'}$ as the state-action representation and the next state representation, respectively, which are random variables as they are outcomes of deterministic mappings of $S, A$ to the representation space. $z_{sa}, z_{s'}$ are the observed instances of $Z_{sa}, Z_{s'}$. Theorem 4.1 states that merely minimizing the Euclidean distance between $Z_{sa}$ and $Z_{s'}$ does not necessarily maximize their mutual information.

**Theorem 4.1.** *Minimizing* $\mathbb{E}[\|Z_{sa} - Z_{s'}\|_2^2]$ *does not necessarily increase the mutual information* $I(Z_{sa}; Z_{s'})$.

The above theorem reveals the pitfalls of prior model-based representation methods, i.e., the trained representations $z_{sa}, z_{s'}$ may fail to capture sufficient information about each other or encode informative knowledge of the latent dynamics by enforcing them to be numerically close, resulting in *biased* representations. In light of this, we deem it necessary to include an additional mutual information loss when learning model-based representations, besides the mean-squared error (MSE) loss, as depicted in Figure 2, i.e.,

$$\mathcal{L} = \min \underbrace{\mathbb{E}[(Z_{sa} - Z_{s'})^2]}_{\text{latent consistency loss}} - \underbrace{\text{MI}(Z_{sa}; Z_{s'})}_{\text{mutual information loss}}. \quad (3)$$

Furthermore, we show in Lemma 4.2 that maximizing the mutual information between $Z_{sa}$ and $Z_{s'}$ can reduce the conditional entropy of $Z_{s'}$ given $Z_{sa}$.

**Lemma 4.2.** *The conditional entropy* $H(Z_{s'}|Z_{sa})$ *strictly reduces if* $I(Z_{s'}; Z_{sa})$ *increases.*

The above lemma is promising since it indicates that the uncertainty of predicting $Z_{s'}$ given $Z_{sa}$ can be effectively reduced by maximizing the mutual information term. This ensures a stronger connection and mapping between the learned representations $z_{sa}$ and $z_{s'}$, and can hopefully benefit the subsequent actor-critic learning. This potential benefit

can be supported by the theoretical insights of DeepMDP (Gelada et al., 2019) and MR.Q (Fujimoto et al., 2025), i.e., the value error is upper-bounded by the transition and reward modeling errors in the latent space, and a more precise latent dynamics directly tightens the bound on the value error. As shown in Lemma 4.2, maximizing $I(Z_{sa}; Z_{s'})$ strictly reduces the conditional entropy $H(Z_{s'}|Z_{sa})$, which implies that the latent dynamics model becomes more deterministic and discriminative when predicting the next state representation. Since we are simultaneously minimizing MSE, the accuracy of the estimated dynamics can be improved, and therefore we can better control the value error upper bound derived in DeepMDP and MR.Q, which ultimately may result in better policy performance.

### 4.2. Faded Prioritized Experience Replay

Another source of bias when learning model-based representations comes from the sampling strategy. Common sampling strategies include uniform sampling and PER (Schaul et al., 2015). Uniform sampling cannot determine whether the transition is worth training. PER compensates this by assigning higher priorities to transitions with larger TD errors. Denote the TD error of the transition $e_i$ in a replay buffer $\mathcal{D}$ as $\delta(i), i \in [0, |\mathcal{D}| - 1], \mathcal{D} = \{e_i\}$. PER follows the sampling probability $p(i) = \frac{|\delta(i)|^\alpha + \kappa}{\sum_j (|\delta(j)|^\alpha + \kappa)}$, where the hyperparameter $\alpha$ smooths out extremes, and $\kappa$ is added to avoid zero probability. We assume $\kappa = 0$ for simplicity, i.e., the sampling probability gives $p(i) = \frac{|\delta(i)|^\alpha}{\sum_j |\delta(j)|^\alpha}$. However, both uniform sampling and PER can suffer from the primacy bias (Nikishin et al., 2022), i.e., overfitting to past experiences, which can deviate far from the distribution of the current policy. Consequently, it may lead to undesired training instability and inferior performance.

Some researchers propose to alleviate the primacy bias by introducing the forget mechanism (Wang et al., 2020; Kang et al., 2025), which focuses more on recent, new experiences and gradually reduces the influence of old experiences with a decay rate $\epsilon, \epsilon \in (0, 1)$. Suppose that the transitions are sequentially added to the replay buffer, with index 0 being the newest transition; the forget mechanism (Kang et al., 2025) generally follows the sampling probability $P(i) = \frac{(1-\epsilon)^i}{\sum_j (1-\epsilon)^j}$. Nevertheless, it is not necessarily true that recent experiences are always worth getting frequently sampled since the new sample may have a small TD error that can contribute less to critic learning. If the transition is less "surprising", it is better to reduce its sampling probability when learning model-based representations.

To enjoy the advantages of the above two types of replay methods and alleviate their negative effects simultaneously, we propose the faded prioritized experience replay (faded PER) strategy, which combines PER and the forget mechanism by assigning high priorities to transitions that are both new and have large TD errors (as shown in Figure 2). To be specific, the faded PER samples transitions via:

$$P(i) = \frac{|\delta(i)|^\alpha (1-\epsilon)^i}{\sum_j |\delta(j)|^\alpha (1-\epsilon)^j}. \quad (4)$$

In this way, the agent can focus more on recent important experiences. As long as the experience is not too old (i.e., deviate from the current policy too much) and its TD error is large, it can still enjoy a comparatively high probability of getting sampled, hence mitigating the negative effects of PER and the forget mechanism. We theoretically analyze the properties of the faded PER in Theorem 4.3.

**Theorem 4.3.** *Let $\mathcal{D}$ be a replay buffer with the decay rate $\epsilon$, $N$ be the batch size, $P(i)$ be the probability that a transition $e_i$ is sampled using faded PER, $\delta(i)$ is the current TD error of $e_i$. Then, we have:*

*(i) for $i_1 < i_2$, if $|\delta(i_1)| = |\delta(i_2)|$, then $P(i_1) > P(i_2)$;*

*(ii) denote $\widehat{P}(i) = \frac{|\delta(i)|^\alpha}{\sum_j |\delta(j)|^\alpha}$, then there exist $C > 0, k \in \mathbb{N}$ such that $\widehat{P}(i)(1-\epsilon)^i \leq P(i) \leq \frac{1}{1+C(1-\epsilon)^{k-i}}$;*

*(iii) the expected sample times of $e_i$, $\mathbb{E}[n_i]$, satisfies $0 < \mathbb{E}[n_i] \leq \frac{N}{1+C(1-\epsilon)^{k-i}} < N, k \in \mathbb{N}, C > 0$.*

The above theorem states that if the two transitions have identical TD errors, the sampling probability of the older experience is strictly lower. Moreover, the sampling probability for any transition $e_i$ under the faded PER can be connected with its sampling probability under PER. Furthermore, the expected sampling times of old experiences are bounded and lie within a constant range. Theorem 4.3 sheds light on adopting the faded PER in practice.

## 4.3. Algorithm

Given the insights above, we propose our empirical algorithm, DR.Q. It mainly debiases existing model-based representation methods from two aspects: (i) incorporates an auxiliary loss for maximizing the mutual information between state-action representations and the next state representation to ensure that the learned representations are sufficiently informative and expressive; (ii) combines PER and the forget mechanism such that most valuable samples are most frequently used while avoiding overfitting to old experiences. The full pseudo-code for DR.Q is deferred to Appendix B.

### 4.3.1. ENCODER TRAINING

The encoders are responsible for modeling the latent dynamics models of the environment, which involve the state encoder $f_\omega(s)$ that outputs the state representation $z_s$, the state-action encoder $g_\omega(z_s, a)$ that produces the state-action representation $z_{sa}$, where $\omega$ is the network parameter, and the linear MDP predictor $M(z_{sa})$ that predicts the next state representation and the reward signal, i.e.,

$$\hat{r}, \hat{z}_{s'} = M(z_{sa}), \quad z_{sa} = g_\omega(z_s, a), \quad z_s = f_\omega(s). \quad (5)$$

where $\hat{z}_{s'}$ is the predicted next state representation. We do not predict the done flags since we empirically find that removing this term has no effect on representation learning or policy learning. The encoder loss of DR.Q is composed of three key terms: the reward loss, the latent dynamics consistency loss, and the mutual information loss.

**Reward loss.** Following MR.Q (Fujimoto et al., 2025), we use a two-hot encoding of the reward, which can be robust to reward magnitude and more effective when dealing with sparse rewards. The locations in the two-hot encoding are determined by the function $\text{symexp}(x) = \text{sign}(x)(\exp(x) - 1)$ (Hafner et al., 2023), resulting in non-uniform intervals. The reward loss is computed based on the cross entropy loss between the two-hot encoding of the true reward $r$ and the predicted reward $\hat{r}$, i.e.,

$$\mathcal{L}_{\text{reward}}(\hat{r}, r) = \text{CE}(\hat{r}, \text{TwoHot}(r)). \quad (6)$$

**Latent dynamics consistency loss.** We fulfill the latent dynamics consistency by minimizing the MSE between the next state representation $\tilde{z}_{s'}$ and the predicted next state representation $\hat{z}_{s'}$, i.e.,

$$\mathcal{L}_{\text{dynamics}}(\hat{z}_{s'}, \tilde{z}_{s'}) = \mathbb{E}[(\hat{z}_{s'} - \text{SG}(\tilde{z}_{s'}))^2], \quad (7)$$

where SG is the stop gradient operator, $\tilde{z}_{s'}$ is produced by the target state encoder network $f_{\omega'}$ parameterized by $\omega'$. We use the target state encoder for better stability. Note that the above loss does not contradict with our previous analysis (i.e., minimize the deviation between $z_{sa}$ and $z_{s'}$) since $\hat{z}_{s'}$

is a linear mapping of the state-action representation $z_{sa}$. Such a mapping is necessary to ensure that the dimensions of $\tilde{z}_{s'}$ and $\hat{z}_{s'}$ are identical.

**Mutual information loss.** This loss is the core component of DR.Q. Unfortunately, it is intractable to directly calculate the mutual information between the next state representation $z_{s'}$ and its predicted counterpart $\hat{z}_{s'}$, especially in high-dimensional spaces. In lieu of infeasible integrals, a common alternative is to optimize the InfoNCE loss (Oord et al., 2018; Poole et al., 2019; Wu et al., 2020; Tschannen et al., 2020; Lu et al., 2023; Chen et al., 2020), which serves as a lower bound of the mutual information, i.e., $I(X;Y) \geq \log(N) - \mathcal{L}_{\text{InfoNCE}}$ for some variables $X, Y$, where $N$ is the sample size. That being said, maximizing the mutual information term is equivalent to minimizing the InfoNCE loss. For any sample $e_i$ in the sampled batch with size $N$, we treat all other samples in the batch as negative samples and measure the cosine similarity between $\hat{z}_{s'}$ and $\tilde{z}_{s'}$. Formally, the InfoNCE loss is computed via:

$$\mathcal{L}_{\text{I}}(\hat{z}_{s'}, \tilde{z}_{s'}) = -\frac{1}{N}\sum_{i=1}^{N}\log\left[\frac{\exp(\cos(\hat{z}_{s'_i}, \tilde{z}_{s'_i})/\tau)}{\sum_{k=1}^{N}\exp(\cos(\hat{z}_{s'_i}, \tilde{z}_{s'_k})/\tau)}\right],$$
(8)

where $\cos(X, Y) = \frac{X \cdot Y}{\|X\|\|Y\|}$ is the cosine similarity measure, $\tau$ is the temperature coefficient.

**Overall encoder loss.** Following prior works (Fujimoto et al., 2025; Hafner et al., 2023; Hansen et al., 2022; 2024), we train the encoders by performing a short-horizon rollout of the learned dynamics model, which has been shown to be effective in quickly adapting to the dynamics of the environment. Specifically, we leverage a subsequence of experience $(s_0, a_0, r_1, s_1, a_1, \ldots, r_H, s_H)$ with a horizon $H$ to generate predictions and representations. The process begins by encoding the initial state $s_0$ followed by the iterative application of the state-action encoder, state encoder, and the MDP predictor (following Equation 5) to produce components like the state-action representation, the predicted reward signal, etc. Consequently, the overall encoder loss of DR.Q is a combination of Equations 6, 7, and 8 summed over the unrolled model,

$$\mathcal{L}_{\text{enc}}^{\text{DR.Q}} = \sum_{t=1}^{H}\lambda_r\mathcal{L}_{\text{reward}}(\hat{r}_t, r_t) + \lambda_d\mathcal{L}_{\text{dynamics}}(\hat{z}_{s',t}, \tilde{z}_{s',t})$$
$$+ \lambda_m\mathcal{L}_{\text{I}}(\hat{z}_{s',t}, \tilde{z}_{s',t}),$$
(9)

where the subscript $t$ denotes the $t$-th step in the horizon, $\lambda_r, \lambda_d, \lambda_m$ are hyperparameters that balance the influence of each component. The target state encoder network $f_{\omega'}$ is periodically updated every $T$ environmental steps.

**Sampling strategy.** Another core component in DR.Q is the faded PER, which combines PER and the forget mechanism to reduce the chances of overfitting to old (bad) experiences

and sampling less important recent experiences. Empirically, we adopt LAP (Fujimoto et al., 2020), an improved version of PER that removes the unnecessary importance sampling ratio terms and sets the minimum priority to be 1. Note that replacing PER with LAP does not affect our theoretical results in Theorem 4.3. Moreover, simply adopting the exponential decay in Equation 4 can quickly decrease the sampling weight of the transition, even if the transition has a large TD error. This can incur a risk of neglecting past valuable transitions. We hence clip the forgetting weight with a threshold $\epsilon_{\text{low}}$. Eventually, for any transition $e_i$, its sampling probability in DR.Q follows:

$$P(i) = \frac{\max(|\delta(i)|^\alpha, 1) \cdot \max(\epsilon_{\text{low}}, (1-\epsilon)^i)}{\sum_j \max(|\delta(j)|^\alpha, 1) \cdot \max(\epsilon_{\text{low}}, (1-\epsilon)^j)}.$$
(10)

### 4.3.2. ACTOR-CRITIC TRAINING

For downstream actor-critic training, we train a deterministic policy network $\pi_\phi$ parameterized by $\phi$ along with its target network $\pi_{\phi'}$, and two critic networks $Q_{\theta_i}(z_{sa})$ in conjunction with their target networks $Q_{\theta'_i}(z_{sa}), i \in \{1, 2\}$.

**Actor.** To enhance the exploration ability of the agent, we add a Gaussian noise $\psi \sim \mathcal{N}(0, \sigma^2)$ to the action:

$$a_\pi = \text{clip}(a', -1, 1), a' = \pi_{\phi'}(z_s) + \text{clip}(\psi, -c, c). \quad (11)$$

The actor objective function then gives:

$$\mathcal{L}_{\text{actor}} = -\frac{1}{2}\sum_{i=1,2}Q_{\theta_i}(z_{sa_\pi}), \quad z_{sa_\pi} = g_\omega(z_s, a_\pi). \quad (12)$$

**Critics.** DR.Q adopts the clipped double Q-learning approach (CDQ) (Fujimoto et al., 2018) to mitigate the function approximation issue. CDQ constructs the target value $y$ by taking the minimum value between the outputs of the two target critic networks. Following prior works (Hessel et al., 2018; Hansen et al., 2022; Fujimoto et al., 2025), we estimate the multi-step return of the critics over a horizon $H_Q$. Furthermore, LAP suggests employing Huber loss instead of the MSE loss to counter the bias from prioritized sampling. Finally, the loss function for critics is given by:

$$\mathcal{L}_{\text{critics}} = \text{Huber}\left(Q_{\theta_i}, \sum_{t=0}^{H_Q-1}\gamma^t r_t + \gamma^{H_Q}\min_{j=1,2}Q_{\theta'_j}\right),$$
(13)

where $Q_{\theta_i} = Q_{\theta_i}(z_{s_0a_0}), i \in \{1, 2\}$, and $\min_j Q_{\theta'_j} = \min_j Q_{\theta'_j}(z_{s_{H_Q}a_{H_Q}}, \pi)$. The target networks of both critics and the actor are periodically copied from their corresponding current networks. Note that DR.Q does not include components such as normalizing target values or input states, parameter reset, regularizing hidden embeddings, etc. We find that the above objective functions are sufficient to achieve strong performance across different benchmarks, making DR.Q simple, clean, and easy to implement.

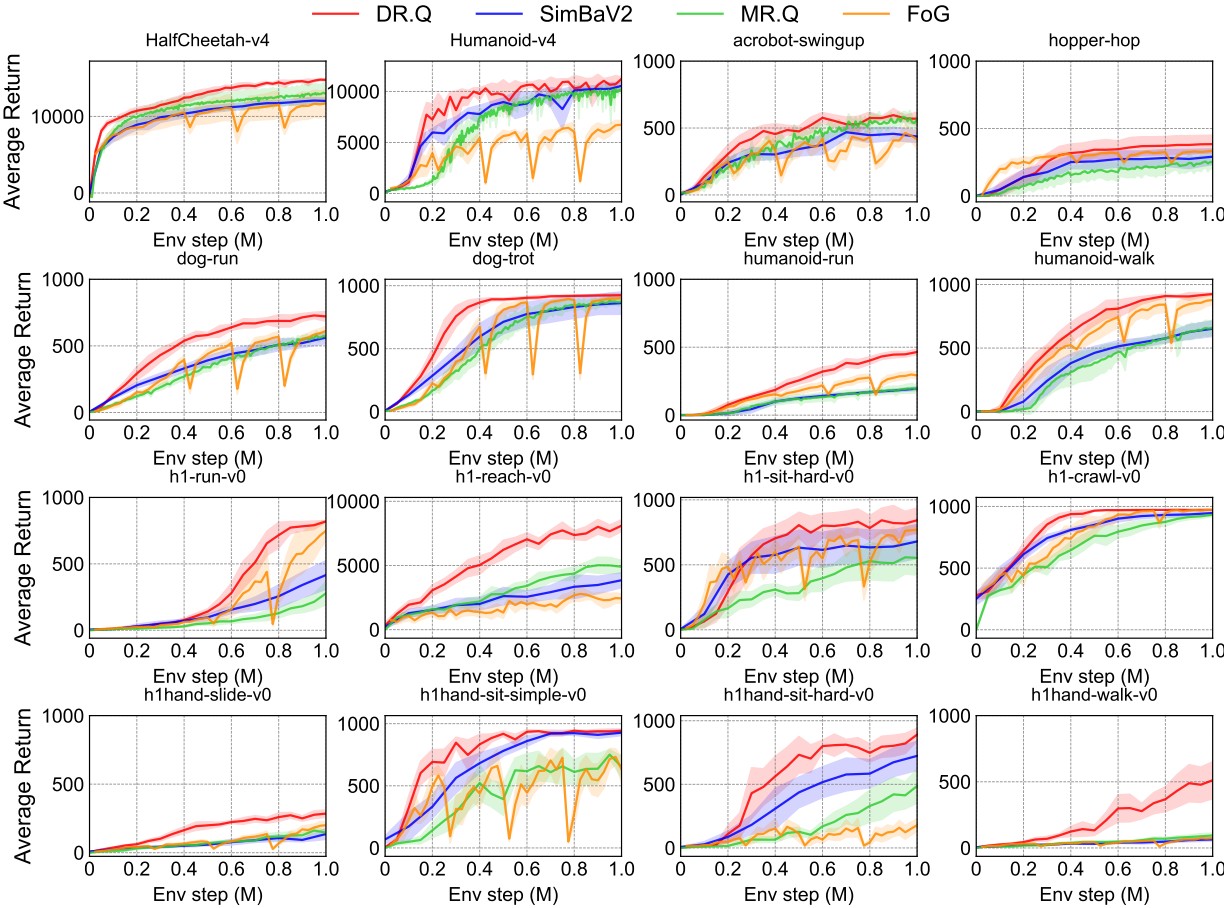

*Figure 3.* **Sample efficiency comparison.** We select 16 representative tasks out of 73 tasks. All results are averaged across 10 random seeds. The solid line denotes the average return and the shaded region indicates the 95% confidence interval.

## 5. Experiment

To comprehensively evaluate the performance of DR.Q, we conduct experiments on three standard continuous control benchmarks, MuJoCo (Todorov et al., 2012), DMC suite (Tassa et al., 2018) (including both proprioceptive control and visual control tasks), and HumanoidBench (Sferrazza et al., 2024), resulting in a total of 73 tasks. These tasks feature varying complexities, spanning from simple, low-dimensional tasks to complex, high-dimensional locomotion tasks. For DMC tasks with vector inputs, we categorize them into DMC-Hard (7 `dog` and `humanoid` tasks) and DMC-Easy (21 tasks). We consider HumanoidBench tasks with or without dexterous hands (14 tasks each).

To better aggregate results across tasks with distinct reward structures, we normalize the performance of agents. MuJoCo results are normalized by TD3 scores (Fujimoto et al., 2018), DMC results are normalized by dividing 1000, and HumanoidBench results are normalized by the task success scores. Agents are trained for 1M steps on MuJoCo tasks, and 500K steps on other tasks (equivalent to 1M frames

in the raw environment due to an action repeat 2). Across all environments and benchmarks, we use a single set of hyperparameters without any algorithmic changes for DR.Q, to show its generality and effectiveness. The detailed hyperparameter setup is deferred to Appendix C.3.

### 5.1. Main Results

**Baselines.** For each benchmark, we consider representative or domain-specific strong baselines for comparison. This covers a wide range of model-free and model-based deep RL algorithms, including PPO (Schulman et al., 2017), DroQ (Hiraoka et al., 2021), TD7 (Fujimoto et al., 2023), REDQ (Chen et al., 2021), DreamerV3 (Hafner et al., 2023), TDMPC2 (Hansen et al., 2024), CrossQ (Bhatt et al., 2024), BRO (Nauman et al., 2024), MAD-TD (Voelcker et al., 2025), SimBa (Lee et al., 2025a), SimBaV2 (Lee et al., 2025b), MR.Q (Fujimoto et al., 2025), FoG (Kang et al., 2025), etc. We report the baseline results from previous works or the original paper if available. For a consistent comparison of sample efficiency and asymptotic performance,

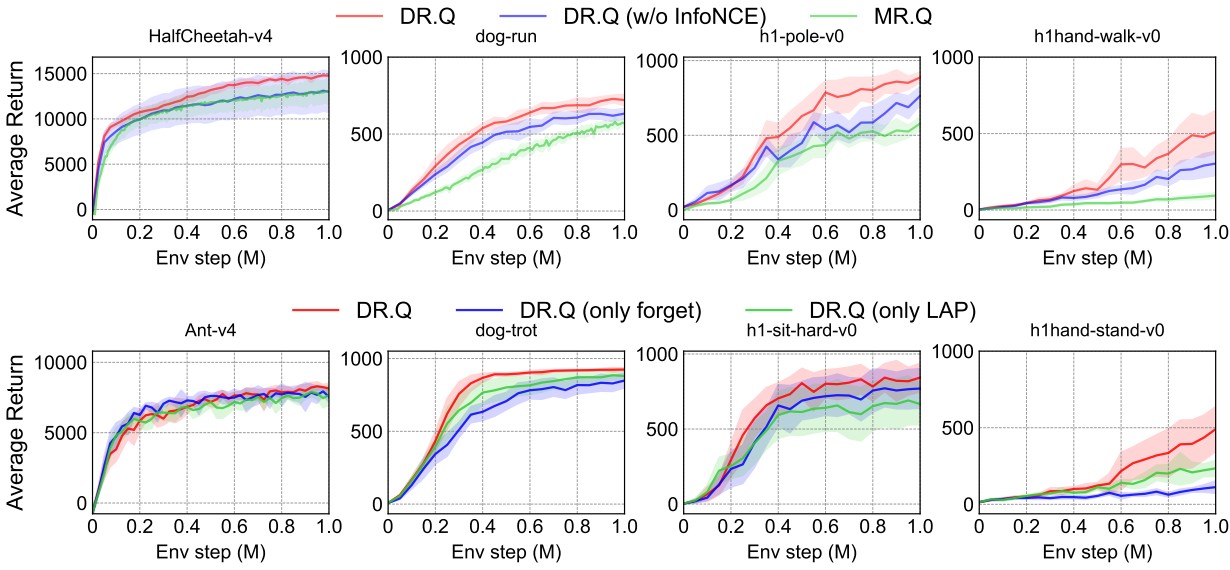

*Figure 4.* **Ablation study** on InfoNCE loss (Top) and sampling strategies (bottom). We adopt 4 representative tasks from different domains. The solid line denotes the average return across 10 seeds, and the shaded regions are the 95% confidence intervals.

we further run representative baselines like MR.Q and Sim-BaV2 on tasks that are not covered in their original papers, using the authors' official code and the suggested default hyperparameters. We evaluate DR.Q across 10 random seeds.

**Performance summary.** To establish a concise and meaningful comparison, we select some leading domain-specific or general algorithms in each benchmark and summarize their normalized average return against DR.Q in Figure 1. DR.Q comprehensively matches or outperforms baseline methods in different domains, often by a substantial margin. Key improvements include a **15.5%** gain over SimBaV2 on DMC-Hard tasks, a **58.9%** improvement against FoG on HumanoidBench (w/ hand) tasks, and a **26.8%** lead over MR.Q on DMC-Visual tasks. Notably, DR.Q consistently outperforms MR.Q across all benchmarks. These results highlight the broad applicability and effectiveness of DR.Q across diverse continuous control tasks. Full results and learning curves of DR.Q are available in Appendix D.

**Sample efficiency.** We summarize in Figure 3 the sample efficiency comparison of DR.Q against strong baselines like MR.Q and SimBaV2. Note that some baselines like BRO are omitted due to a lack of logged results on partial tasks. We deem it acceptable, as the included baselines have been shown to exhibit higher sample efficiency than the omitted ones. As depicted, DR.Q achieves remarkably better sample efficiency on numerous challenging tasks, sometimes surpassing baselines by a large margin. To the best of our knowledge, DR.Q is the first that achieves an average return that exceeds 700 on the challenging `dog-run` task under 1M environment steps.

### 5.2. Ablation Study

In this part, we investigate the design choices of DR.Q by conducting experiments on selected tasks. Results in wider environments can be found in Appendix E.1, where we also ablate the necessity of the latent dynamics consistency loss. We further show the effectiveness of the mutual information loss and provide representation visualization results of DR.Q and MR.Q in Appendix E.3.

**The InfoNCE loss.** One of the key designs in DR.Q is the InfoNCE loss, which ensures that the learned representations can better encode the latent dynamics knowledge. To evaluate its effectiveness, we exclude its contribution by setting $\lambda_m = 0$. Results in Figure 4 (top) show that removing the InfoNCE loss generally incurs inferior performance, especially on some high-dimensional HumanoidBench tasks where the state space may contain redundant information (e.g., dexterous hands), making the agent struggle to extract useful patterns from the proprioceptive input. Since DR.Q enjoys a similar way of learning model-based representations as MR.Q, its performance is at least as competitive as MR.Q, even without the InfoNCE loss term.

**Faded PER.** Another important component in DR.Q is the faded PER, which assigns sampling probability based on both the TD error and the sampling weight decay. To demonstrate its importance, we consider the following variants of DR.Q, DR.Q (only forget) that merely adopts the forget mechanism for replaying experiences, and DR.Q (only LAP) which removes the forget mechanism. Since the superiority of the forget mechanism over (corrected) uniform sampling (Yenicesu et al., 2024) is already established in

prior work (Kang et al., 2025), we exclude these baselines from our comparison. Empirical results in Figure 4 (bottom) show that either excluding PER or the forget mechanism can result in a risk of degrading the performance and sample efficiency of the agent, while combining them can bring maximum performance gains across all evaluated tasks.

## 6. Conclusion

This paper proposes DR.Q, a general off-policy RL algorithm that trains model-based representations for mastering diverse continuous control tasks with a single set of hyperparameters. DR.Q debiases existing model-based representation methods by (i) introducing the InfoNCE loss to maximize the mutual information between the state-action representation and the next state representation, apart from minimizing their deviations; (ii) ensuring that recent and important transitions are more frequently sampled by combining the prioritized experience replay approach and the forget mechanism. Empirical results across 73 continuous control tasks from three standard benchmarks show that DR.Q achieves competitive or better performance compared to prior strong model-free or model-based baselines. DR.Q provides a concrete step towards building high-performing and general model-free algorithms. For future work, one can evaluate DR.Q on discrete action tasks like Atari, find better paradigms for learning model-based representations, or build a stronger model-free RL algorithm based on DR.Q.

**Limitations.** Despite the strong performance of DR.Q on numerous challenging tasks, its performance on tasks like `Hopper-v4` is inferior, which can be a side effect of adopting unified hyperparameters across all benchmarks. DR.Q also fails on challenging visual DMC tasks like `humanoid-run`, while we notice that all methods fail to achieve meaningful scores on this task. DrQv2 requires 15M environment steps to achieve meaningful performance on `visual-humanoid-run`, while we only run DR.Q and baselines for 1M steps. The encoders may not capture good representations for downstream policy and critic learning within such a limited budget. DR.Q introduces InfoNCE loss and Faded PER, which may introduce extra computation overhead, but it remains more efficient than baselines like SimBaV2 or FoG because it maintains a Replay Ratio (UTD) of 1. Faded PER only requires storing a 1D forget weight array. InfoNCE computation is minimal since the batch size and representation dimensions are not large. Akin to MR.Q, DR.Q is not designed for hard exploration tasks or non-Markovian tasks, and it may fail on those tasks. Furthermore, we do not evaluate DR.Q on discrete action tasks like Atari, since we set our focus on continuous control tasks and it is very expensive to run those experiments.

## Acknowledgments

This work was supported by NSFC in part under Grant 62450001 and 62476008. The authors would like to thank the anonymous reviewers for their valuable comments and advice.

## Impact Statement

This paper presents work whose goal is to advance the field of Machine Learning. There are many potential societal consequences of our work, none of which we feel must be specifically highlighted here.

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

# A. Missing Proofs

**Theorem A.1.** *Minimizing $\mathbb{E}[\|Z_{sa} - Z_{s'}\|_2^2]$ does not necessarily increase the mutual information $I(Z_{sa}; Z_{s'})$.*

*Proof.* We use the method of contradiction to prove this theorem. We construct two simple toy examples to show that minimizing the MSE term $\mathbb{E}[\|Z_{sa} - Z_{s'}\|_2^2]$ can either minimize the mutual information $I(Z_{sa}; Z_{s'})$ or increase it.

For simplicity, we first consider $Z_{sa} = X, Z_{s'} = X + \epsilon$, where $X \sim \mathcal{N}(\mathbf{0}, I_d), \epsilon \sim \mathcal{N}(\mathbf{0}, \sigma^2 I_d)$, where the random variable $\epsilon$ is independent of the random variable $X$, $I_d$ is the identity matrix with rank $d$. Then, it is easy to find that

$$\mathbb{E}[\|Z_{sa} - Z_{s'}\|_2^2] = \mathbb{E}[\|X - (X + \epsilon)\|_2^2] = \mathbb{E}[\|\epsilon\|_2^2] = \sigma^2 d. \tag{14}$$

The mutual information gives

$$I(Z_{sa}; Z_{s'}) = I(X; X + \epsilon) = H(X + \epsilon) - H(X + \epsilon | X) = H(X + \epsilon) - H(\epsilon). \tag{15}$$

Note that $X$ and $\epsilon$ are independent, we then have $X + \epsilon \sim \mathcal{N}(\mathbf{0}, (1 + \sigma^2)I_d)$, and hence,

$$I(Z_{sa}; Z_{s'}) = H(X + \epsilon) - H(\epsilon) = \frac{d}{2} \log(2\pi e(1 + \sigma^2)) - \frac{d}{2} \log(2\pi e \sigma^2) = \frac{d}{2} \log\left(1 + \frac{1}{\sigma^2}\right). \tag{16}$$

Observing Equation 14 and Equation 16, we conclude that when $\sigma^2$ becomes large, the MSE term becomes large while the mutual information term becomes small, and vise versa.

Nevertheless, if we let $Z_{sa} = X, Z_{s'} = kX$, where $k > 1, k \in \mathbb{R}, X \sim \mathcal{N}(\mathbf{0}, I_d)$, then we have $Z_{s'} = kX \sim \mathcal{N}(\mathbf{0}, k^2 I_d)$. Following the same procedure, we have the MSE term,

$$\mathbb{E}[\|Z_{sa} - Z_{s'}\|_2^2] = \mathbb{E}[\|X - kX\|_2^2] = \mathbb{E}[\|(1 - k)X\|_2^2] = (1 - k)^2 d. \tag{17}$$

For the mutual information term, we can derive that

$$I(Z_{sa}; Z_{s'}) = H(kX) - H(kX | X) = H(kX) = \frac{d}{2} \log(2\pi e k^2). \tag{18}$$

Observing Equation 17 and Equation 18, we conclude that when $k$ approaches 1, the MSE term becomes small while the mutual information term also becomes small ($\log(2\pi e k^2) \to 0$), while when $k \to \infty$, both the MSE term and the mutual information term become large.

We can now conclude that there is no definite correlation between the MSE term and the mutual information term. That being said, minimizing $\mathbb{E}[\|Z_{sa} - Z_{s'}\|_2^2]$ does not necessarily ensure that the mutual information $I(Z_{sa}; Z_{s'})$ increases. $\square$

**Remark:** Note that $X$ does not necessarily have a mean of 0. The above two examples are selected for simplicity and the benefit of theoretical analysis. Since the conclusion does not hold on two simple examples, it does not hold in a general case. Intuitively, minimizing the MSE loss only ensures that the Euclidean distance between the state-action representation $Z_{sa}$ and the state representation $Z_{s'}$ becomes small, while not directly optimizing their distributions. It is possible that their MSE loss is small while their distributions differ.

**Lemma A.2.** *$H(Z_{s'} | Z_{sa})$ strictly reduces if $I(Z_{s'}; Z_{sa})$ increases.*

*Proof.* This conclusion is quite straightforward. By the definition of the mutual information, we have

$$H(Z_{s'} | Z_{sa}) = H(Z_{s'}) - I(Z_{s'}; Z_{sa}), \tag{19}$$

We note that the state entropy term $H(Z_{s'})$ is the inherent property of the environment and would not change during algorithmic optimization. It is then clear that the conditional entropy $H(Z_{s'} | Z_{sa})$ gets strictly smaller when the mutual information term $I(Z_{s'}; Z_{sa})$ becomes large. $\square$

**Theorem A.3.** *Let $\mathcal{D}$ be a replay buffer with the decay rate $\epsilon$, $N$ be the batch size, $P(i)$ be the probability that a transition $e_i$ is sampled using faded prioritized experience replay, $\delta(i)$ is the current TD error of $e_i$. Then, we have:*

*(i) for $i_1 < i_2$, if $|\delta(i_1)| = |\delta(i_2)|$, then $P(i_1) > P(i_2)$;*

*(ii) denote $\widehat{P}(i) = \frac{|\delta(i)|^\alpha}{\sum_j |\delta(j)|^\alpha}$, then there exist $C > 0, k \in \mathbb{N}$ such that $\widehat{P}(i)(1 - \epsilon)^i \le P(i) \le \frac{1}{1 + C(1 - \epsilon)^{k-i}}$;*

*(iii) the expected sample times of $e_i$, $\mathbb{E}[n_i]$, satisfies $0 < \mathbb{E}[n_i] \le \frac{N}{1 + C(1 - \epsilon)^{k-i}} < N, k \in \mathbb{N}, C > 0$.*

*Proof.* According to the faded prioritized experience replay, the probability that the transition $e_i$ gets sampled is given by:

$$P(i) = \frac{|\delta(i)|^\alpha (1 - \epsilon)^i}{\sum_{j=0}^{|\mathcal{D}|-1} |\delta(j)|^\alpha (1 - \epsilon)^j}. \tag{20}$$

For (i), since $|\delta(i_1)| = |\delta(i_2)|$, we have,

$$P(i_1) - P(i_2) = \frac{|\delta(i_1)|^\alpha (1 - \epsilon)^{i_1} - |\delta(i_2)|^\alpha (1 - \epsilon)^{i_2}}{\sum_j |\delta(j)|^\alpha (1 - \epsilon)^j} = \frac{|\delta(i_1)|^\alpha [(1 - \epsilon)^{i_1} - (1 - \epsilon)^{i_2}]}{\sum_j |\delta(j)|^\alpha (1 - \epsilon)^j} > 0. \tag{21}$$

For (ii), it is easy to find that,

$$P(i) = \frac{|\delta(i)|^\alpha (1 - \epsilon)^i}{\sum_{j=0}^{|\mathcal{D}|-1} |\delta(j)|^\alpha (1 - \epsilon)^j} \ge \frac{|\delta(i)|^\alpha (1 - \epsilon)^i}{\sum_{j=0}^{|\mathcal{D}|-1} |\delta(j)|^\alpha} = \frac{|\delta(i)|^\alpha}{\sum_{j=0}^{|\mathcal{D}|-1} |\delta(j)|^\alpha} (1 - \epsilon)^i = \widehat{P}(i)(1 - \epsilon)^i. \tag{22}$$

For the upper bound, we have $\sum_{j=0}^{|\mathcal{D}|} |\delta(j)|^\alpha (1 - \epsilon)^j = |\delta(0)|^\alpha (1 - \epsilon)^0 + |\delta(1)|^\alpha (1 - \epsilon)^1 + \ldots + |\delta(i)|^\alpha (1 - \epsilon)^i + \ldots + |\delta(k)|^\alpha (1 - \epsilon)^k + \ldots + |\delta(|\mathcal{D}| - 1)|^\alpha (1 - \epsilon)^{|\mathcal{D}|-1}$, then there exists $k \in \mathbb{N}$ such that $|\delta(k)| > 0$, which incurs,

$$\sum_{j=0}^{|\mathcal{D}|} |\delta(j)|^\alpha (1 - \epsilon)^j \ge |\delta(i)|^\alpha (1 - \epsilon)^i + |\delta(k)|^\alpha (1 - \epsilon)^k. \tag{23}$$

This indicates that

$$P(i) = \frac{|\delta(i)|^\alpha (1 - \epsilon)^i}{\sum_{j=0}^{|\mathcal{D}|-1} |\delta(j)|^\alpha (1 - \epsilon)^j} \le \frac{|\delta(i)|^\alpha (1 - \epsilon)^i}{|\delta(i)|^\alpha (1 - \epsilon)^i + |\delta(k)|^\alpha (1 - \epsilon)^k}. \tag{24}$$

If $|\delta(i)| > 0$, then $|\delta(i)|^\alpha > 0$, and,

$$P(i) \le \frac{|\delta(i)|^\alpha (1 - \epsilon)^i}{|\delta(i)|^\alpha (1 - \epsilon)^i + |\delta(k)|^\alpha (1 - \epsilon)^k} = \frac{1}{1 + \frac{|\delta(k)|^\alpha}{|\delta(i)|^\alpha} (1 - \epsilon)^{k-i}}. \tag{25}$$

Since $|\delta(i)| > 0, |\delta(k)| > 0$, there must exists $C \in \mathbb{R}, C > 0$ such that $\frac{|\delta(k)|^\alpha}{|\delta(i)|^\alpha} \ge C$, and hence there exists $C > 0$ such that

$$P(i) \le \frac{|\delta(i)|^\alpha (1 - \epsilon)^i}{|\delta(i)|^\alpha (1 - \epsilon)^i + |\delta(k)|^\alpha (1 - \epsilon)^k} = \frac{1}{1 + \frac{|\delta(k)|^\alpha}{|\delta(i)|^\alpha} (1 - \epsilon)^{k-i}} \le \frac{1}{1 + C(1 - \epsilon)^{k-i}}. \tag{26}$$

If $|\delta(i)| = 0$, then $P(i) = 0 < \frac{1}{1 + C(1 - \epsilon)^{k-i}}$ also holds.

For (iii), we have $\widehat{P}(i) = \frac{|\delta(i)|^\alpha}{\sum_j |\delta(j)|^\alpha} > 0$, and hence

$$\mathbb{E}[n_i] = N \times P(i) \ge N \times \widehat{P}(i) \times (1 - \epsilon)^i > 0. \tag{27}$$

Furthermore, based on (ii), there exists $k \in \mathbb{N}, C > 0$ such that $P(i) \le \frac{1}{1 + C(1 - \epsilon)^{k-i}}$, hence for any transition $e_i$, its expected sample times can then be bounded,

$$\mathbb{E}[n_i] = N \times P(i) \le N \frac{1}{1 + C(1 - \epsilon)^{k-i}} = \frac{N}{1 + C(1 - \epsilon)^{k-i}} < N. \tag{28}$$

$\square$

# B. Pseudo-code of DR.Q

---

**Algorithm 1** Debiased model-based Representations for Q-learning (DR.Q)

---

**Input:** Faded PER forget probability threshold $\epsilon_{\text{low}}$, loss weights $\lambda_d, \lambda_r, \lambda_m$, multi-step return horizon $H_Q$, encoder rollout horizon $H$, LAP smoothing coefficient $\alpha$, temperature coefficient $\tau$, exploration step $T_{\text{explore}}$, target update frequency $T_{\text{target}}$, noise clip threshold $c$

1: Initialize policy network $\pi_\phi$, critic networks $Q_{\theta_1}, Q_{\theta_2}$, the linear MDP predictor $M(\cdot)$, the state encoder $f_\omega$ and state-action encoder $g_\omega$ with random parameters
2: Initialize target networks $\phi' \leftarrow \phi, \theta'_1 \leftarrow \theta_1, \theta'_2 \leftarrow \theta_2, \omega' \leftarrow \omega$ and empty replay buffer $\mathcal{B} = \{\}$
3: **for** $t = 1$ to $T$ **do**
4:     Select action $a$ via Equation 11, i.e., $a = \text{clip}(a', -1, 1), a' = \pi_{\phi'}(z_s) + \text{clip}(\psi, -c, c), z_s = f_\omega(s), \psi \sim \mathcal{N}(0, 0.2^2)$.
5:     Execute action $a$ and observe reward $r$, new state $s'$ and done flag $d$
6:     Store transitions in the replay buffer, i.e., $\mathcal{B} \leftarrow \mathcal{B} \bigcup \{(s, a, r, s', d)\}$
7:     **if** $t > T_{\text{explore}}$ **then**
8:         // Encoder Training
9:         **if** $t \% T_{\text{target}} = 0$ **then**
10:             Update target networks: $\theta'_1, \theta'_2, \phi', \omega' \leftarrow \theta_1, \theta_2, \phi, \omega$
11:             **for** $T_{\text{target}}$ time steps **do**
12:                 Sample transitions via Faded PER (Equation 10) and update the encoders via Equation 9, i.e., $\mathcal{L}_{\text{enc}}^{\text{DR.Q}} = \sum_{t=1}^{H} \lambda_r \mathcal{L}_{\text{reward}}(\hat{r}_t, r_t) + \lambda_d \mathcal{L}_{\text{dynamics}}(\hat{z}_{s',t}, \tilde{z}_{s',t}) + \lambda_m \mathcal{L}_{\text{I}}(\hat{z}_{s',t}, \tilde{z}_{s',t})$, with $\mathcal{L}_{\text{reward}}, \mathcal{L}_{\text{dynamics}}, \mathcal{L}_{\text{I}}$ defined in Equations 6, 7, 8, respectively
13:             **end for**
14:         **end if**
15:         // Critic Training
16:         Sample transitions via Faded PER (Equation 10) and update critic networks via Equation 13, i.e., $\mathcal{L}_{\text{critics}} = \text{Huber}\left(Q_{\theta_i}, \sum_{t=0}^{H_Q-1} \gamma^t r_t + \gamma^{H_Q} \min_{j=1,2} Q_{\theta'_j}\right)$
17:         // Actor Training
18:         Update the actor network via Equation 12, i.e., $\mathcal{L}_{\text{actor}} = -\frac{1}{2} \sum_{i=1,2} Q_{\theta_i}(z_{sa_\pi}), z_{sa_\pi} = g_\omega(z_s, a_\pi), a_\pi \sim \pi_\phi(z_s)$.
19:         // Update Priority
20:         Update the LAP priority with the TD errors
21:     **end if**
22: **end for**

---

# C. Experiment Setup

In this section, we present the benchmark details and experiment setup for running DR.Q and baseline methods.

## C.1. Environment Details

**Gym MuJoCo** (Brockman et al., 2016) is a suite of locomotion tasks that are widely used in the context of RL research. We choose the five most commonly adopted environments by prior works (Fujimoto et al., 2023; 2025; Lee et al., 2025b) and employ the v4 version of these tasks. To better aggregate scores, we follow MR.Q (Fujimoto et al., 2025) and normalize the returns with the random scores and TD3 (Fujimoto et al., 2018) scores:

$$\text{TD3-Normalized}(x) := \frac{x - \text{random score}}{\text{TD3 score} - \text{random score}}. \tag{29}$$

We run all algorithms on the MuJoCo tasks for 1M environment steps with no action repeat and summarize the random scores and TD3 scores on each environment in Table 1.

**DMC suite** (Tassa et al., 2018) is a standard continuous control benchmark, which collects numerous locomotion and manipulation tasks with varying complexities, spanning from low-dimensional tasks to complex, high-dimensional locomotion tasks. For proprioceptive DMC tasks (i.e., vector inputs), we consider 28 tasks and divide them into two categories: DMC-Easy, which involves 21 tasks, and DMC-Hard, which involves 4 dog tasks and 3 humanoid tasks. For visual control,

*Table 1.* **The complete list of Gym MuJoCo tasks.** Obs denotes observation.

| Task | Random | TD3 | Obs dim $|\mathcal{O}|$ | Action dim $|\mathcal{A}|$ |
|---|---|---|---|---|
| Ant-v4 | $-70.288$ | 3942 | 27 | 8 |
| HalfCheetah-v4 | $-289.415$ | 10574 | 17 | 6 |
| Hopper-v4 | 18.791 | 3226 | 11 | 3 |
| Humanoid-v4 | 120.423 | 5165 | 376 | 17 |
| Walker2d-v4 | 2.791 | 3946 | 17 | 6 |

the state is made up of the previous 3 observations which are resized to $84 \times 84$ pixels in RGB format, as MR.Q (Fujimoto et al., 2025) does. For either proprioceptive or visual control, we use an action repeat of 2. We run 500K steps for all algorithms (1M environment steps due to action repeat) and report the average return in each environment directly. We provide the full list of evaluated DMC tasks (with vector inputs) in Table 2.

*Table 2.* **The full list of DMC suite tasks.** Obs represents observation.

| Task | Obs dim $|\mathcal{O}|$ | Action dim $|\mathcal{A}|$ |
|---|---|---|
| acrobot-swingup | 6 | 1 |
| ball-in-cup-catch | 6 | 1 |
| cartpole-balance | 5 | 1 |
| cartpole-balance-sparse | 5 | 1 |
| cartpole-swingup | 5 | 1 |
| cartpole-swingup-sparse | 5 | 1 |
| cheetah-run | 17 | 6 |
| dog-run | 223 | 38 |
| dog-trot | 223 | 38 |
| dog-stand | 223 | 38 |
| dog-walk | 223 | 38 |
| finger-spin | 9 | 2 |
| finger-turn-easy | 12 | 2 |
| finger-turn-hard | 12 | 2 |
| fish-swim | 24 | 5 |
| hopper-hop | 15 | 4 |
| hopper-stand | 15 | 4 |
| humanoid-run | 67 | 24 |
| humanoid-stand | 67 | 24 |
| humanoid-walk | 67 | 24 |
| pendulum-swingup | 3 | 1 |
| quadruped-run | 78 | 12 |
| quadruped-walk | 78 | 12 |
| reacher-easy | 6 | 2 |
| reacher-hard | 6 | 2 |
| walker-run | 24 | 6 |
| walker-stand | 24 | 6 |
| walker-walk | 24 | 6 |

**HumanoidBench** (Sferrazza et al., 2024) is a high-dimensional simulated robot learning benchmark. It employs the Unitree H1 humanoid robot and is designed to evaluate the performance of the humanoid robot across a variety of challenging whole-body manipulation and locomotion tasks. The benchmark provides tasks that either contain dexterous hands or do not. If one chooses to involve the dexterous hands, the observation space and the action space of the environment will become much larger. Previous methods like SimBaV2 (Lee et al., 2025b) mainly focus on tasks without dexterous hands, while we include tasks with or without dexterous hands simultaneously to comprehensively evaluate the performance and sample efficiency of the agent under high-dimensional tasks. We adopt 14 default locomotion tasks without dexterous hands in the

SimBa paper (Lee et al., 2025a) and an additional 14 locomotion tasks with hands. For all tasks, we run baselines and DR.Q for 500 steps with an action repeat 2, which is equivalent to 1M environment steps. Since HumanoidBench tasks also have varied reward scales across different environments, we normalize the undiscounted return results using the random scores and the task success scores provided in the SimBaV2 paper (Lee et al., 2025b):

$$\text{Success-Normalized}(x) := \frac{x - \text{random score}}{\text{Task success score} - \text{random score}}. \tag{30}$$

We summarize the evaluated tasks on HumanoidBench and other necessary information in Table 3.

*Table 3.* **A full list of HumanoidBench tasks.** Obs means observation. Random means the return of the random policy, and Success means the task success return. We use the same task success scores for tasks with or without dexterous hands.

| Task | Random | Success | Obs dim $|\mathcal{O}|$ | Action dim $|\mathcal{A}|$ |
|---|---|---|---|---|
| h1-balance-hard | 9.044 | 800 | 77 | 19 |
| h1-balance-simple | 9.391 | 800 | 64 | 19 |
| h1-crawl | 272.658 | 700 | 51 | 19 |
| h1-hurdle | 2.214 | 700 | 51 | 19 |
| h1-maze | 106.441 | 1200 | 51 | 19 |
| h1-pole | 20.09 | 700 | 51 | 19 |
| h1-reach | 260.302 | 12000 | 57 | 19 |
| h1-run | 2.02 | 700 | 51 | 19 |
| h1-sit-simple | 9.393 | 750 | 51 | 19 |
| h1-sit-hard | 2.448 | 750 | 64 | 19 |
| h1-slide | 3.191 | 700 | 51 | 19 |
| h1-stair | 3.112 | 700 | 51 | 19 |
| h1-stand | 10.545 | 800 | 51 | 19 |
| h1-walk | 2.377 | 700 | 51 | 19 |
| h1hand-basketball | 8.979 | 1200 | 164 | 61 |
| h1hand-bookshelf-simple | 16.777 | 2000 | 308 | 61 |
| h1hand-bookshelf-hard | 14.848 | 2000 | 308 | 61 |
| h1hand-crawl | 278.868 | 700 | 151 | 61 |
| h1hand-door | 2.771 | 600 | 155 | 61 |
| h1hand-pole | 19.721 | 700 | 151 | 61 |
| h1hand-reach | −50.024 | 12000 | 157 | 61 |
| h1hand-run | 1.927 | 700 | 151 | 61 |
| h1hand-slide | 3.142 | 700 | 151 | 61 |
| h1hand-sit-simple | 10.768 | 750 | 151 | 61 |
| h1hand-sit-hard | 2.477 | 750 | 164 | 61 |
| h1hand-stair | 3.161 | 700 | 151 | 61 |
| h1hand-stand | 11.973 | 800 | 151 | 61 |
| h1hand-walk | 2.505 | 700 | 151 | 61 |

## C.2. Baselines

**PPO** (Schulman et al., 2017). Proximal Policy Optimization (PPO) is a classical and widely used general-purpose on-policy model-free RL algorithm. Results of PPO on Gym MuJoCo tasks, DMC-Hard tasks, and DMC-Visual tasks are obtained from the MR.Q paper (Fujimoto et al., 2025), which are averaged over 10 seeds.

**TD3+OFE** (Ota et al., 2020). It proposes an online feature extractor network (OFENet) that trains the state-action representation and the next state representation by enforcing the latent dynamics consistency. Eventually, TD3+OFE achieves strong performance over TD3 (Fujimoto et al., 2018). Results of TD3+OFE on Gym MuJoCo tasks are obtained from the TD7 paper (Fujimoto et al., 2023), which are averaged across 10 seeds.

**TQC** (Kuznetsov et al., 2020). Truncated Quantile Critic (TQC) controls the overestimation bias by using distributional critics and truncating the atoms with highest value estimates. Results of TQC on MuJoCo tasks are obtained from the TD7 paper (Fujimoto et al., 2023). These scores are averaged over 10 seeds.

**REDQ** (Chen et al., 2021). Randomized Ensembled Double Q-Learning (REDQ) boots sample efficiency by using a large update-to-data (UTD) ratio and an ensemble of Q-functions. For MuJoCo tasks, the results of REDQ are obtained from the CrossQ paper (Bhatt et al., 2024) with 10 seeds and an UTD ratio 20.

**DroQ** (Hiraoka et al., 2021). Dropout Q-Function (DroQ) reduces the computational overhead of REDQ by using dropout and layer normalization. For MuJoCo tasks, the results of DroQ are averaged across 10 seeds and an UTD ratio 20. The results are obtained from the CrossQ paper (Bhatt et al., 2024).

**DreamerV3** (Hafner et al., 2023). DreamerV3 is a general purpose model-based RL algorithm that learns a world model in a compact latent space. For MuJoCo tasks and DMC tasks, we use results from MR.Q (Fujimoto et al., 2025), which are averaged over 10 seeds. For HumanoidBench (w/o hand) tasks, we use results from SimBa (Lee et al., 2025a), which are averaged over 3 seeds. The results of HumanoidBench (w/ hand) tasks are obtained from the HumanoidBench authors (https://github.com/carlosferrazza/humanoid-bench/tree/main/logs) across 3 seeds.

**DrQ-v2** (Yarats et al., 2021a). DrQ-v2 is built upon DrQ (Yarats et al., 2021b) with several refinements: (i) replacing the base algorithm from SAC (Haarnoja et al., 2018) to DDPG (Lillicrap et al., 2015); (ii) incorporating multi-step returns; (iii) adding bilinear interpolation to the random shift image augmentation; (iv) introducing an exploration schedule; (v) finding better hyperparameters. Its results in DMC-Visual tasks are taken directly from the MR.Q paper (Fujimoto et al., 2025), which are averaged across 10 seeds.

**TD7** (Fujimoto et al., 2023). TD7 leverages latent consistency loss for training state-action representation and state-representation, and feed them forward along with raw states and actions to learn value functions and the policy. It further adopts policy checkpoints and prioritized experience replay. Results of MuJoCo and DMC tasks are taken directly from MR.Q (Fujimoto et al., 2025), which are averaged across 10 seeds. Results on HumanoidBench (w/o hand) tasks are obtained from SimBaV2 (Lee et al., 2025b), which are averaged over 5 seeds.

**TDMPC2** (Hansen et al., 2024). TDMPC2 is a model-based RL algorithm that learns a latent dynamics model and performs model predictive control by using the learned model. Its average performance across 10 seeds on MuJoCo and DMC tasks are obtained from MR.Q (Fujimoto et al., 2025). The results on HumanoidBench (w/o hand) tasks are obtained from SimBa paper (Lee et al., 2025a), and its results on HumanoidBench (w/ hand) tasks are obtained from the HumanoidBench authors (https://github.com/carlosferrazza/humanoid-bench/tree/main/logs), all using 3 random seeds.

**CrossQ** (Bhatt et al., 2024). CrossQ is an algorithm that achieves high sample efficiency with low replay ratio by removing target networks and using careful batch normalization. The results on Gym MuJoCo tasks are obtained from the SimBaV2 paper (Lee et al., 2025b), which are averaged across 10 seeds.

**iQRL** (Scannell et al., 2024a). Implicitly Quantized Reinforcement Learning (iQRL) improves the sample efficiency of the agent via latent quantization. The results on DMC-Hard tasks are obtained from its original paper, averaged across 3 seeds.

**BRO** (Nauman et al., 2024). BRO improves the sample efficiency of the agent by scaling the critic networks in conjunction with some regularization techniques. It adopts distributional Q-learning, optimistic exploration with an additional exploration policy, and periodic resets. For Gym MuJoCo tasks, DMC-Easy tasks, and HumanoidBench (w/o hand) tasks, the results are averaged across 5 seeds. For DMC-Hard tasks, the results are averaged across 10 seeds. All results are obtained from the SimBa paper (Lee et al., 2025a) with an UTD ratio 2.

**MAD-TD** (Voelcker et al., 2025). Model-Augmented Data for Temporal Difference learning (MAD-TD) stabilizes the high UTD training by adding a small fraction of $\alpha$ model-generated synthetic data. Its results averaged across 10 seeds on DMC-Hard tasks are taken directly from its original paper with an UTD = 8 and $\alpha = 0.05$.

**MR.Q** (Fujimoto et al., 2025). Model-based Representations for Q-learning (MR.Q) is a general-purpose model-free RL algorithm that leverages model-based objectives (e.g., latent dynamics consistency, reward predictions) for learning state-action representations and state representations. It achieves strong performance across numerous benchmarks with a single set of hyperparameters. Results on Gym MuJoCo, DMC-Easy, DMC-Hard and DMC-Visual tasks are obtained directly from the MR.Q paper, which are averaged across 10 seeds. For HumanoidBench tasks with or without dexterous hands (28 tasks), we run MR.Q on them with the official codebase (https://github.com/facebookresearch/MRQ) and summarize the results across 10 random seeds. Note that MR.Q sets the replay ratio to be 1.

**SimBa** (Lee et al., 2025a). SimBa is a new network architecture designed for scaling the agent. It involves several tricks and design choices like state normalization, residual blocks, and layer normalization. For DMC-Hard tasks, the results are averaged across 15 seeds. On Gym MuJoCo, DMC-Easy and HumanoidBench (w/o hand) tasks, we use 10 seeds. These scores are obtained directly from the SimBaV2 paper (Lee et al., 2025b). For HumanoidBench (w/ hand) tasks, we obtain results across 10 random seeds using the official codebase (https://github.com/SonyResearch/simba) with an UTD ratio 2.

**SimBaV2** (Lee et al., 2025b). SimBaV2 is an improved network architecture built upon the SimBa architecture. It stabilizes optimization through two key mechanisms: first, constraining the growth of weight and feature norms via hyperspherical normalization; and second, employing reward-scaled distributional value estimation to maintain stable gradients across varying reward magnitudes. Its results on Gym MuJoCo tasks, DMC-Easy tasks, DMC-Hard tasks, and HumanoidBench (w/o hand) tasks are taken directly from its original paper, which are averaged across 10 seeds. For HumanoidBench (w/ hand) tasks, we run the official codebase (https://github.com/DAVIAN-Robotics/SimBaV2) with an UTD ratio 2 and 10 random seeds.

**FoG** (Kang et al., 2025). Forget and Grow (FoG) employs two key components, forgetting early experiences to balance memory, and dynamically expanding the network to better exploit the patterns of existing data. It utilizes the Offline Boosted Actor-Critic (Luo et al., 2024) as the backbone algorithm, and adopts tricks like periodic resets. Results on all benchmarks are acquired by running the authors' official codebase (https://github.com/nothingbutbut/FoG) with an UTD ratio 10. We summarize the results across 10 seeds.

Despite some papers claiming that they adopt a single set of hyperparameters for different tasks, it turns out that many methods are not actually general-purpose, e.g., FoG, SimBaV2. Some of them set different algorithmic configurations for different tasks or adopt different hyperparameters. We summarize the comparison of DR.Q against some representative and strong baselines below.

*Table 4.* **Comparison of DR.Q against baseline methods.** CDQ denotes the clipped double Q-learning, and AvgQ is the average Q-learning that take mean of two critics.

| Algorithm | Hyperparameters | Algorithmic Configurations | Details |
|---|---|---|---|
| DR.Q | Fixed | Fixed | N/A |
| PPO | Fixed | Fixed | N/A |
| MR.Q | Fixed | Fixed | N/A |
| TDMPC2 | Fixed | Fixed | N/A |
| SimBa | Fixed | Not Fixed | CDQ on HumanoidBench, single Q otherwise |
| SimBaV2 | Fixed | Not Fixed | CDQ on HumanoidBench, single Q otherwise |
| FoG | Not Fixed | Not Fixed | different batch size and reset steps on HumanoidBench; CDQ or AvgQ on different tasks |

## C.3. Hyperparameters

We summarize the hyperparameters adopted in DR.Q in Table 5, which are kept fixed across all benchmarks. Note that DR.Q also keeps its algorithmic configurations unchanged across all tasks.

*Table 5.* **DR.Q hyperparameters.** We keep all these hyperparameters fixed across all benchmark tasks.

| Hyperparameter | Value |
|---|---|
| Target policy noise $\sigma$ | $\mathcal{N}(0, 0.2^2)$ |
| Target policy noise clipping $c$ | $(-0.3, 0.3)$ |
| LAP probability smoothing $\alpha$ | 0.4 |
| LAP minimum priority | 1 |
| Exploration steps | $10^4$ |
| Exploration noise | $\mathcal{N}(0, 0.2^2)$ |
| Discount factor $\gamma$ | 0.99 |
| Replay buffer size | $10^6$ |
| Batch size | 256 |
| Target update frequency $T_{\text{target}}$ | 250 |
| Replay ratio (UTD) | 1 |
| Optimizer (all networks) | AdamW (Loshchilov & Hutter, 2019) |
| Weight initialization (all networks) | Xavier uniform (Glorot & Bengio, 2010) |
| Bias initialization (all networks) | 0 |
| Encoder learning rate | $3 \times 10^{-4}$ |
| Encoder weight decay | 0.01 |
| Encoder $z_s$ dim | 512 |
| Encoder $z_{sa}$ dim | 512 |
| Encoder $z_a$ dim (used only in the architecture) | 256 |
| Encoder hidden dimension | 750 |
| Encoder activation function | ELU (Clevert et al., 2015) |
| Encoder reward bins | 65 |
| Encoder reward range | $[-10, 10]$ |
| Encoder horizon $H$ | 5 |
| Actor learning rate | $3 \times 10^{-4}$ |
| Actor hidden dim | 512 |
| Actor activation function | ReLU |
| Critic learning rate | $3 \times 10^{-4}$ |
| Critic hidden dim | 512 |
| Critic activation function | ELU (Clevert et al., 2015) |
| Critic multi-step return horizon $H_Q$ | 3 |
| Latent consistency loss weight $\lambda_d$ | 1 |
| Reward loss weight $\lambda_r$ | 0.1 |
| InfoNCE loss weight $\lambda_m$ | 0.1 |
| Faded PER decay rate $\epsilon$ | 0.0001 |
| Faded PER forget weight threshold $\epsilon_{\text{low}}$ | 0.1 |

# D. Full Main Results

In this section, we provide complete and thorough comparison of DR.Q against extended prior domain-specific or general algorithms. We also provide learning curves of DR.Q against some typical algorithms on each domain. We focus on MR.Q (Fujimoto et al., 2025), SimBaV2 (Lee et al., 2025b), TDMPC2 (Hansen et al., 2024), and FoG (Kang et al., 2025) as main baselines. These methods are selected due to their strong performance across numerous challenging benchmarks. The learning curves of some algorithms are omitted since the authors do not provide raw logs or result files.

The shaded areas in the figures and the gray bracketed terms in the tables denote 95% bootstrapped confidence interval. Following SimBaV2 (Lee et al., 2025b), we compute the 95% bootstrapped confidence interval for each task across $n$ random seeds via:

$$\text{CI} = \left[\mu - 1.96 \times \frac{\sigma}{\sqrt{n}}, \mu + 1.96 \times \frac{\sigma}{\sqrt{n}}\right],$$

where $\mu, \sigma$ are the sample mean and the sample standard deviation, respectively. For aggregated mean, median, interquartile mean (IQM), confidence intervals are computed over $n \times T$ samples using rliable[1] (Agarwal et al., 2021), where $T$ is the number of evaluated tasks in the benchmark.

---

[1]https://github.com/google-research/rliable

## D.1. Gym MuJoCo Results

As shown in Table 6, DR.Q outperforms SimBaV2 in 3 out of 5 environments, though the average normalized score of DR.Q slightly lags behind that of SimBaV2. This is mainly due to the poor performance on Hopper-v4, where other recent strong methods like MR.Q and FoG also fail to achieve strong performance.

*Table 6.* **Full comparison results on Gym MuJoCo tasks.** We include a wide range of domain-specific or general model-free and model-based RL algorithms for comparison. We report the final average performance at 1M environment step for each task. The [brackets] denote a 95% bootstrap confidence interval. The aggregate mean, median and interquartile mean (IQM) are computed over the TD3-normalized score as described in Appendix C.1.

| Task | TD7 | TDMPC2 | MR.Q | FoG | SimBaV2 | DR.Q |
|---|---|---|---|---|---|---|
| Ant-v4 | 8509 [8168, 8844] | 4751 [2988, 6145] | 6901 [6261, 7482] | 6761 [6161, 7360] | 7429 [7209, 7649] | 8138 [7764, 8511] |
| HalfCheetah-v4 | 17433 [17301, 17559] | 15078 [14065, 15932] | 12939 [11663, 13762] | 11709 [9928, 13491] | 12022 [11640, 12404] | 14775 [14638, 14912] |
| Hopper-v4 | 3511 [3236, 3736] | 2081 [1197, 2921] | 2692 [2131, 3309] | 1822 [1316, 2327] | 4054 [3929, 4179] | 2504 [1931, 3077] |
| Humanoid-v4 | 7428 [7304, 7553] | 6071 [5770, 6333] | 10223 [9929, 10498] | 6737 [6319, 7155] | 10546 [10195, 10897] | 11239 [11052, 11426] |
| Walker2d-v4 | 6096 [5621, 6547] | 3008 [1706, 4321] | 6039 [5644, 6386] | 5124 [4719, 5529] | 6938 [6691, 7185] | 6422 [5123, 7721] |
| IQM | 1.540 [1.500, 1.580] | 1.050 [0.890, 1.190] | 1.499 [1.361, 1.650] | 1.242 [1.117, 1.349] | 1.637 [1.470, 1.791] | 1.691 [1.473, 1.879] |
| Median | 1.550 [1.450, 1.630] | 1.180 [0.830, 1.220] | 1.488 [1.340, 1.623] | 1.261 [1.080, 1.344] | 1.616 [1.49, 1.744] | 1.564 [1.416, 1.806] |
| Mean | 1.570 [1.540, 1.600] | 1.040 [0.920, 1.150] | 1.465 [1.346, 1.585] | 1.196 [1.082, 1.307] | 1.617 [1.513, 1.718] | 1.608 [1.449, 1.759] |

| Task | PPO | DreamerV3 | SimBa | TD3+OFE | TQC | REDQ | DroQ | CrossQ | BRO |
|---|---|---|---|---|---|---|---|---|---|
| Ant-v4 | 1584 [1360, 1815] | 1947 [1076, 2813] | 5882 [5354, 6411] | 7398 [7280, 7516] | 3582 [2489, 4675] | 5314 [4539, 6090] | 5965 [5560, 6370] | 6980 [6834, 7126] | 7027 [6710, 7343] |
| HalfCheetah-v4 | 1744 [1523, 2118] | 5502 [3717, 7123] | 9422 [8745, 10100] | 13758 [13214, 14302] | 12349 [11471, 13227] | 11505 [10213, 12798] | 11070 [10272, 11867] | 12893 [11771, 14015] | 13747 [12621, 14873] |
| Hopper-v4 | 3022 [2633, 3339] | 2666 [2106, 3210] | 3231 [3004, 3458] | 3121 [2615, 3627] | 3526 [3302, 3750] | 3299 [2730, 3869] | 2797 [2387, 3208] | 2467 [1855, 3079] | 2122 [1655, 2588] |
| Humanoid-v4 | 477 [436, 518] | 4217 [2785, 5523] | 6513 [5634, 7392] | 6032 [5698, 6366] | 6029 [5498, 6560] | 5278 [5127, 5430] | 5380 [5353, 5407] | 10480 [10307, 10653] | 4757 [3139, 6376] |
| Walker2d-v4 | 2487 [1907, 3022] | 4519 [3692, 5244] | 4290 [3864, 4716] | 5195 [4683, 5707] | 5321 [4999, 5643] | 5228 [4836, 5620] | 4781 [4539, 5024] | 6257 [5277, 7237] | 3432 [2064, 4801] |
| IQM | 0.410 [0.110, 0.834] | 0.720 [0.620, 0.850] | 1.114 [1.043, 1.200] | 1.261 [1.035, 1.680] | 1.143 [0.971, 1.290] | 1.135 [1.086, 1.194] | 1.108 [1.055, 1.170] | 1.565 [1.394, 1.710] | 1.071 [0.828, 1.333] |
| Median | 0.412 [0.071, 0.936] | 0.810 [0.580, 0.930] | 1.143 [1.063, 1.227] | 1.293 [0.967, 1.861] | 1.163 [0.910, 1.349] | 1.188 [1.086, 1.241] | 1.133 [1.068, 1.199] | 1.489 [1.317, 1.643] | 1.071 [0.884, 1.322] |
| Mean | 0.447 [0.186, 0.725] | 0.760 [0.670, 0.860] | 1.147 [1.075, 1.223] | 1.322 [1.090, 1.615] | 1.137 [1.012, 1.261] | 1.160 [1.096, 1.224] | 1.134 [1.067, 1.205] | 1.475 [1.330, 1.608] | 1.101 [0.927, 1.278] |

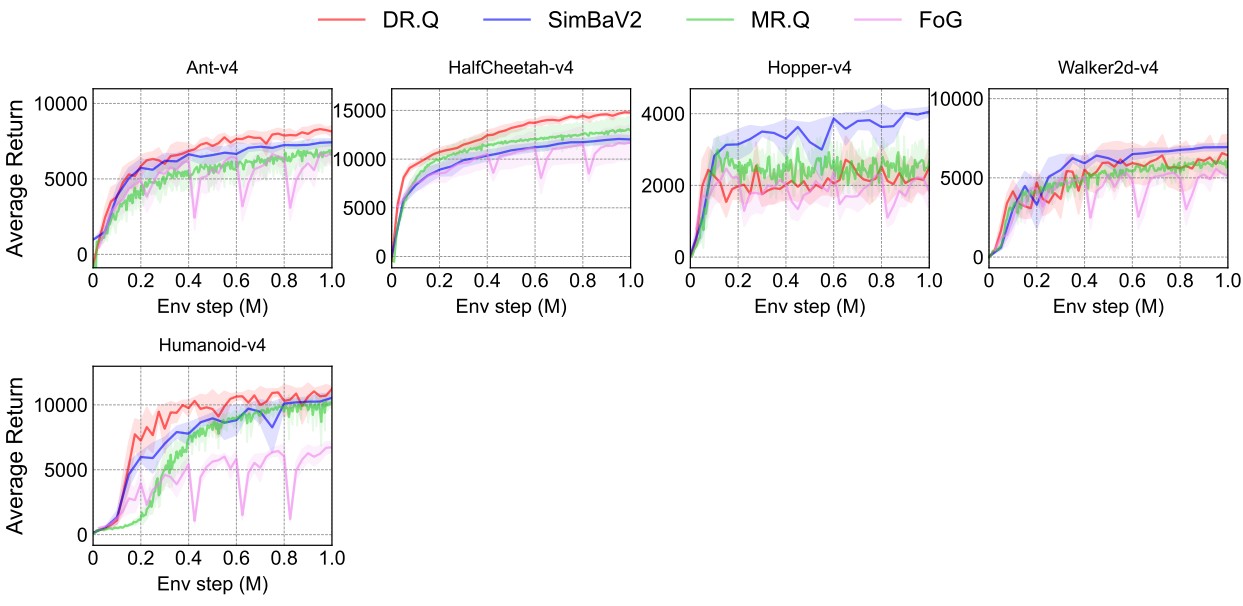

*Figure 5.* **Full learning curves on Gym MuJoCo tasks.** We report the average episode return results across 10 random seeds. The shaded region denotes the 95% bootstrap confidence intervals.

## D.2. DMC Suite Easy Results

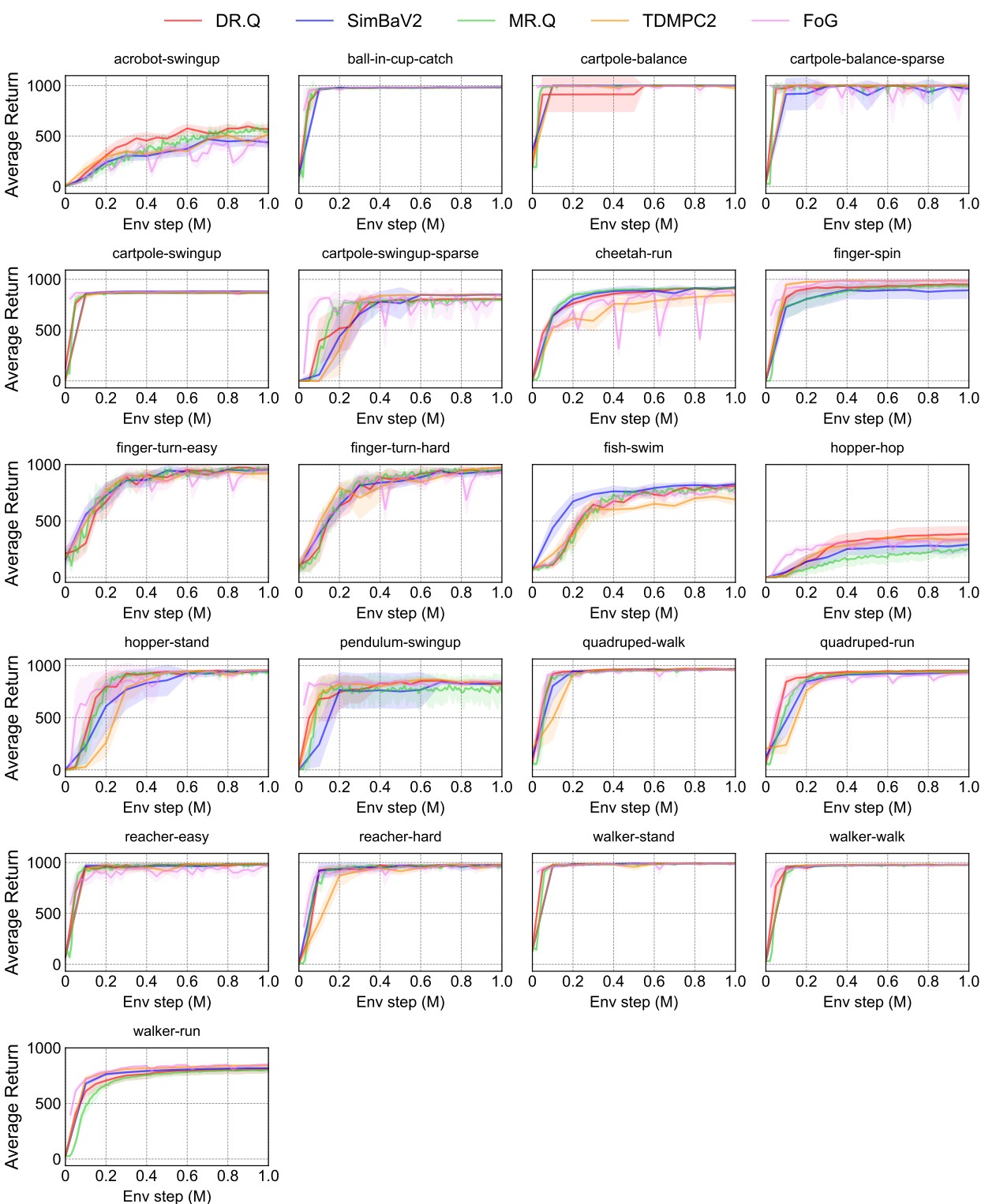

*Figure 6.* **Full learning curves on DMC suite easy tasks.** The solid lines denote the average return in each environment and the shaded regions denote the 95% bootstrap confidence intervals.

*Table 7.* **Full performance comparison results in DMC-Easy tasks.** We report the final average performance at 500K steps (1M environment steps due to the action repeat 2). The values in [brackets] is a 95% bootstrap confidence interval. The aggregate mean, median and interquartile mean (IQM) are reported in units of 1k.

| Task | MR.Q | Simba | SimbaV2 | FoG | DR.Q |
|---|---|---|---|---|---|
| acrobot-swingup | 567 [523, 616] | 431 [379, 482] | 436 [391, 482] | 414 [344, 485] | 569 [519, 619] |
| ball-in-cup-catch | 981 [979, 984] | 981 [978, 983] | 982 [980, 984] | 983 [981, 985] | 980 [979, 982] |
| cartpole-balance | 999 [999, 1000] | 998 [998, 999] | 999 [999, 999] | 997 [996, 999] | 999 [999, 1000] |
| cartpole-balance-sparse | 1000 [1000, 1000] | 991 [973, 1008] | 967 [904, 1030] | 1000 [1000, 1000] | 987 [963, 1012] |
| cartpole-swingup | 866 [866, 866] | 876 [871, 881] | 880 [876, 883] | 881 [880, 882] | 867 [866, 867] |
| cartpole-swingup-sparse | 798 [780, 818] | 825 [795, 854] | 848 [848, 849] | 840 [829, 850] | 805 [791, 818] |
| cheetah-run | 877 [849, 905] | 920 [918, 922] | 821 [642, 913] | 838 [732, 944] | 911 [905, 918] |
| finger-spin | 937 [917, 956] | 849 [758, 939] | 891 [810, 972] | 987 [986, 989] | 949 [917, 980] |
| finger-turn-easy | 953 [931, 974] | 935 [903, 968] | 953 [925, 980] | 949 [920, 977] | 956 [932, 980] |
| finger-turn-hard | 950 [910, 974] | 915 [859, 972] | 951 [925, 977] | 921 [863, 978] | 949 [923, 975] |
| fish-swim | 792 [773, 810] | 823 [799, 846] | 826 [806, 846] | 744 [701, 786] | 808 [788, 828] |
| hopper-hop | 251 [195, 301] | 385 [322, 449] | 290 [233, 348] | 335 [326, 345] | 384 [317, 451] |
| hopper-stand | 951 [948, 955] | 929 [900, 957] | 944 [926, 962] | 956 [953, 959] | 954 [949, 959] |
| pendulum-swingup | 748 [597, 829] | 737 [575, 899] | 827 [805, 849] | 838 [810, 866] | 835 [819, 852] |
| quadruped-run | 947 [940, 954] | 928 [916, 939] | 935 [928, 943] | 918 [906, 929] | 953 [949, 957] |
| quadruped-walk | 963 [959, 967] | 957 [951, 963] | 962 [955, 969] | 963 [960, 966] | 969 [964, 973] |
| reacher-easy | 983 [983, 985] | 983 [981, 986] | 983 [979, 986] | 980 [971, 990] | 975 [958, 993] |
| reacher-hard | 977 [975, 980] | 966 [947, 984] | 967 [946, 987] | 965 [944, 986] | 976 [973, 979] |
| walker-run | 793 [765, 815] | 796 [792, 801] | 817 [812, 821] | 851 [848, 853] | 809 [775, 844] |
| walker-stand | 988 [987, 990] | 985 [982, 989] | 987 [984, 990] | 987 [985, 989] | 991 [989, 992] |
| walker-walk | 978 [978, 980] | 975 [972, 978] | 976 [974, 978] | 978 [977, 980] | 979 [976, 982] |
| IQM | 0.936 [0.917, 0.952] | 0.922 [0.905, 0.938] | 0.933 [0.918, 0.948] | 0.935 [0.919, 0.951] | 0.937 [0.920, 0.951] |
| Median | 0.876 [0.847, 0.905] | 0.870 [0.841, 0.896] | 0.875 [0.847, 0.905] | 0.874 [0.845, 0.904] | 0.885 [0.863, 0.912] |
| Mean | 0.874 [0.848, 0.898] | 0.864 [0.84, 0.887] | 0.874 [0.849, 0.897] | 0.873 [0.847, 0.897] | 0.886 [0.865, 0.906] |

| Task | DreamerV3 | TD7 | TDMPC2 | BRO | DR.Q |
|---|---|---|---|---|---|
| acrobot-swingup | 230 [193, 266] | 58 [38, 75] | 584 [551, 615] | 529 [504, 555] | 569 [519, 619] |
| ball-in-cup-catch | 968 [965, 973] | 984 [982, 986] | 983 [981, 985] | 982 [981, 984] | 980 [979, 982] |
| cartpole-balance | 998 [997, 1000] | 999 [998, 1000] | 996 [995, 998] | 999 [998,999] | 999 [999, 1000] |
| cartpole-balance-sparse | 1000 [1000, 1000] | 999 [1000, 1000] | 1000 [1000, 1000] | 852 [563, 1141] | 987 [963, 1012] |
| cartpole-swingup | 736 [591, 838] | 869 [866, 873] | 875 [870, 880] | 879 [877, 882] | 867 [866, 867] |
| cartpole-swingup-sparse | 702 [560, 792] | 573 [333, 806] | 845 [839, 849] | 840 [827, 852] | 805 [791, 818] |
| cheetah-run | 917 [915, 920] | 699 [655, 744] | 914 [911, 917] | 863 [822, 904] | 911 [905, 918] |
| finger-spin | 666 [577, 763] | 335 [99, 596] | 986 [986, 988] | 988 [987, 989] | 949 [917, 980] |
| finger-turn-easy | 906 [883, 927] | 912 [774, 983] | 979 [975, 983] | 957 [923, 992] | 956 [932, 980] |
| finger-turn-hard | 864 [812, 900] | 470 [199, 727] | 947 [916, 977] | 957 [920, 993] | 949 [923, 975] |
| fish-swim | 813 [808, 819] | 86 [64, 120] | 659 [615, 706] | 618 [523, 713] | 808 [788, 828] |
| hopper-hop | 116 [66, 165] | 87 [25, 160] | 425 [368, 500] | 295 [273, 316] | 384 [317, 451] |
| hopper-stand | 747 [669, 806] | 670 [466, 829] | 952 [944, 958] | 949 [941, 957] | 954 [949, 959] |
| pendulum-swingup | 774 [740, 802] | 500 [251, 743] | 846 [830, 862] | 829 [795, 864] | 835 [819, 852] |
| quadruped-run | 130 [92, 169] | 645 [567, 713] | 942 [938, 947] | 859 [824, 895] | 953 [949, 957] |
| quadruped-walk | 193 [137, 243] | 949 [939, 957] | 963 [959, 967] | 958 [949, 967] | 969 [964, 973] |
| reacher-easy | 966 [964, 970] | 970 [951, 982] | 983 [980, 986] | 983 [983, 984] | 975 [958, 993] |
| reacher-hard | 919 [864, 955] | 898 [861, 936] | 960 [936, 979] | 974 [970, 978] | 976 [973, 979] |
| walker-run | 510 [430, 588] | 804 [783, 825] | 854 [851, 859] | 790 [776, 805] | 809 [775, 844] |
| walker-stand | 941 [934, 948] | 983 [974, 989] | 991 [990, 994] | 990 [986, 994] | 991 [989, 992] |
| walker-walk | 898 [875, 919] | 977 [975, 980] | 981 [979, 984] | 979 [975, 983] | 979 [976, 982] |
| IQM | 0.813 [0.621, 0.899] | 0.771 [0.570, 0.907] | 0.941 [0.880, 0.973] | 0.928 [0.899, 0.952] | 0.937 [0.920, 0.951] |
| Median | 0.813 [0.702, 0.917] | 0.804 [0.573, 0.949] | 0.952 [0.875, 0.981] | 0.872 [0.819, 0.906] | 0.885 [0.863, 0.912] |
| Mean | 0.714 [0.584, 0.832] | 0.689 [0.548, 0.816] | 0.889 [0.819, 0.946] | 0.861 [0.823, 0.896] | 0.886 [0.865, 0.906] |

## D.3. DMC Suite Hard Results

*Table 8.* **Full performance comparison on DMC-Hard tasks.** We report the final average return results at 500K steps (1M environment step due to action repeat 2). The [bracketed values] represent a 95% bootstrap confidence interval. The aggregate mean, median and interquartile mean (IQM) are reported in units of 1k.

| Task | TDMPC2 | MR.Q | Simba | SimBaV2 | FoG | DR.Q |
|---|---|---|---|---|---|---|
| dog-run | 265 [166, 342] | 569 [547, 595] | 544 [525, 564] | 562 [516, 608] | 613 [577, 648] | 721 [684, 758] |
| dog-stand | 506 [266, 715] | 967 [960, 975] | 960 [951, 969] | 981 [977, 985] | 976 [969, 982] | 972 [963, 982] |
| dog-trot | 407 [265, 530] | 877 [845, 898] | 824 [773, 876] | 861 [772, 950] | 901 [892, 911] | 925 [914, 936] |
| dog-walk | 486 [240, 704] | 916 [908, 924] | 916 [905, 928] | 935 [927, 944] | 921 [909, 933] | 950 [942, 958] |
| humanoid-run | 181 [121, 231] | 200 [170, 236] | 181 [171, 191] | 194 [182, 207] | 292 [268, 317] | 465 [444, 485] |
| humanoid-stand | 658 [506, 745] | 868 [822, 903] | 846 [801, 890] | 916 [886, 945] | 931 [921, 941] | 938 [932, 944] |
| humanoid-walk | 754 [725, 791] | 662 [610, 724] | 668 [608, 728] | 651 [590, 713] | 878 [839, 917] | 925 [918, 932] |
| IQM | 0.464 [0.305, 0.632] | 0.796 [0.724, 0.860] | 0.773 [0.713, 0.83] | 0.808 [0.726, 0.879] | 0.880 [0.818, 0.914] | 0.917 [0.871, 0.936] |
| Median | 0.486 [0.265, 0.658] | 0.722 [0.654, 0.797] | 0.706 [0.647, 0.772] | 0.729 [0.655, 0.808] | 0.788 [0.724, 0.855] | 0.844 [0.796, 0.893] |
| Mean | 0.465 [0.329, 0.606] | 0.723 [0.660, 0.781] | 0.706 [0.656, 0.755] | 0.729 [0.664, 0.791] | 0.787 [0.730, 0.840] | 0.842 [0.800, 0.881] |

| Task | DreamerV3 | TD7 | PPO | iQRL | BRO | MAD-TD |
|---|---|---|---|---|---|---|
| dog-run | 4 [4, 5] | 69 [36, 101] | 26 [26, 28] | 380 [336, 424] | 374 [338, 411] | 437 [396, 478] |
| dog-stand | 22 [20, 27] | 582 [432, 741] | 129 [122, 139] | 926 [897, 955] | 966 [956, 977] | 967 [952, 982] |
| dog-trot | 10 [6, 17] | 21 [13, 30] | 31 [30, 34] | 713 [516, 909] | 783 [717, 848] | 867 [805, 929] |
| dog-walk | 17 [15, 21] | 52 [19, 116] | 40 [37, 43] | 866 [827, 905] | 931 [920, 942] | 924 [906, 943] |
| humanoid-run | 0 [1, 1] | 57 [23, 92] | 0 [1, 1] | 188 [167, 210] | 204 [186, 223] | 200 [186, 220] |
| humanoid-stand | 5 [5, 6] | 317 [117, 516] | 5 [5, 6] | 727 [655, 799] | 920 [909, 931] | 870 [840, 901] |
| humanoid-walk | 1 [1, 2] | 176 [42, 320] | 1 [1, 2] | 688 [642, 735] | 672 [619, 725] | 684 [609, 759] |
| IQM | 0.008 [0.002, 0.016] | 0.134 [0.047, 0.343] | 0.021 [0.003, 0.069] | 0.694 [0.528, 0.805] | 0.772 [0.662, 0.854] | 0.787 [0.691, 0.865] |
| Median | 0.005 [0.001, 0.018] | 0.069 [0.052, 0.317] | 0.026 [0.001, 0.040] | 0.640 [0.516, 0.766] | 0.694 [0.615, 0.774] | 0.707 [0.634, 0.786] |
| Mean | 0.009 [0.003, 0.015] | 0.182 [0.062, 0.336] | 0.033 [0.009, 0.068] | 0.642 [0.531, 0.747] | 0.693 [0.625, 0.757] | 0.708 [0.642, 0.771] |

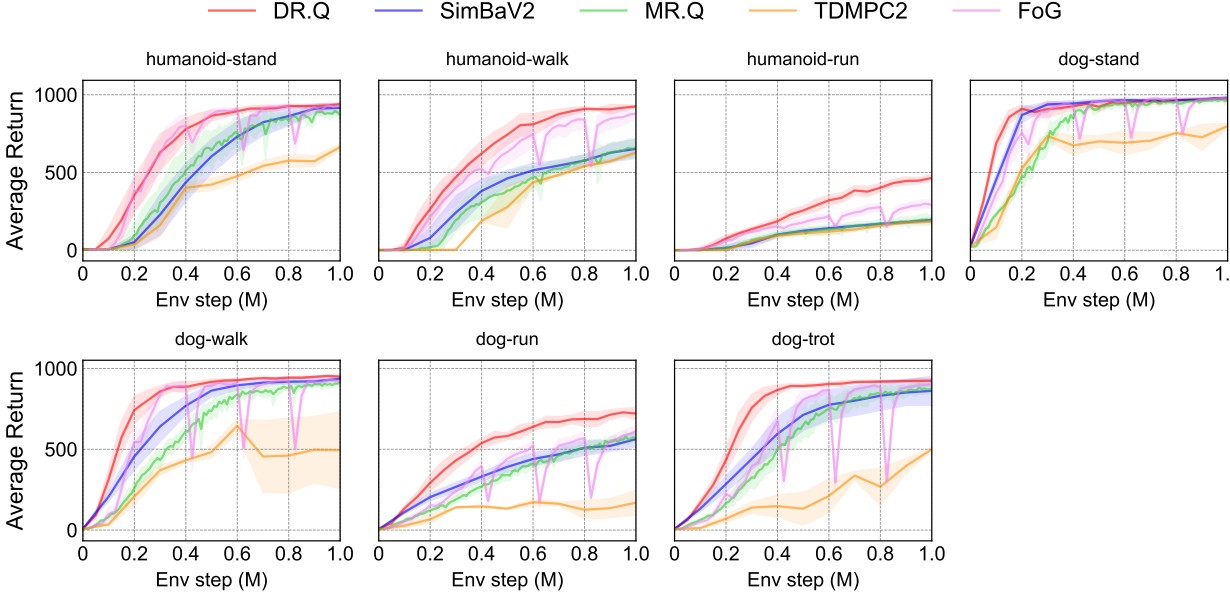

*Figure 7.* **Full learning curves on DMC suite hard tasks.** The solid lines denote the average return in each environment and the shaded regions denote the 95% bootstrap confidence intervals.

## D.4. HumanoidBench (w/o Hand) Results

*Table 9.* **Full comparison results on HumanoidBench without dexterous hands.** We report the final average return results at 1M environment steps (equivalent to 500K steps due to an action repeat 2). The [bracketed values] denote a 95% bootstrap confidence interval. The aggregate mean, median and interquartile mean (IQM) are computed over the success normalized score as described in Appendix C.1.

| Task | Simba | SimbaV2 | MR.Q | FoG | DR.Q |
|---|---|---|---|---|---|
| h1-pole-v0 | 716 [667, 765] | 791 [785, 797] | 578 [534, 623] | 893 [846, 940] | 887 [853, 921] |
| h1-slide-v0 | 277 [252, 303] | 487 [404, 571] | 303 [270, 337] | 674 [562, 785] | 355 [324, 386] |
| h1-stair-v0 | 269 [153, 385] | 493 [467, 518] | 235 [213, 257] | 466 [383, 548] | 401 [328, 475] |
| h1-balance-hard-v0 | 75 [71, 80] | 143 [128, 157] | 69 [67, 72] | 81 [71, 91] | 92 [87, 97] |
| h1-balance-simple-v0 | 337 [193, 482] | 723 [651, 795] | 135 [110, 160] | 616 [536, 696] | 205 [166, 244] |
| h1-sit-hard-v0 | 512 [354, 670] | 679 [548, 811] | 553 [421, 686] | 770 [738, 802] | 843 [747, 939] |
| h1-sit-simple-v0 | 833 [814, 853] | 875 [870, 880] | 850 [819, 882] | 828 [800, 856] | 931 [924, 938] |
| h1-maze-v0 | 354 [342, 366] | 313 [287, 340] | 344 [340, 347] | 331 [310, 353] | 354 [349, 359] |
| h1-crawl-v0 | 923 [904, 942] | 946 [933, 959] | 932 [919, 945] | 971 [969, 973] | 973 [972, 974] |
| h1-hurdle-v0 | 175 [150, 201] | 202 [167, 236] | 131 [108, 155] | 114 [100, 129] | 344 [245, 443] |
| h1-reach-v0 | 3874 [3220, 4527] | 3850 [3272, 4427] | 4902 [4390, 5414] | 2434 [2083, 2785] | 8101 [7640, 8563] |
| h1-run-v0 | 232 [185, 279] | 415 [307, 524] | 278 [192, 364] | 749 [666, 832] | 820 [815, 824] |
| h1-stand-v0 | 772 [701, 843] | 814 [770, 857] | 800 [754, 846] | 671 [516, 825] | 856 [815, 897] |
| h1-walk-v0 | 550 [391, 709] | 845 [840, 850] | 716 [657, 775] | 866 [859, 872] | 850 [830, 869] |
| IQM | 0.521 [0.413, 0.633] | 0.799 [0.686, 0.908] | 0.519 [0.417, 0.630] | 0.846 [0.713, 0.969] | 0.864 [0.735, 0.976] |
| Median | 0.598 [0.514, 0.692] | 0.781 [0.693, 0.865] | 0.602 [0.516, 0.687] | 0.794 [0.705, 0.899] | 0.823 [0.733, 0.920] |
| Mean | 0.606 [0.536, 0.678] | 0.776 [0.705, 0.849] | 0.604 [0.531, 0.677] | 0.802 [0.721, 0.883] | 0.825 [0.748, 0.902] |

| Task | DreamerV3 | TD7 | TDMPC2 | DR.Q |
|---|---|---|---|---|
| h1-pole-v0 | 41 [28, 54] | 441 [320, 563] | 744 [609, 879] | 887 [853, 921] |
| h1-slide-v0 | 11 [7, 15] | 39 [26, 53] | 334 [304, 364] | 355 [324, 386] |
| h1-stair-v0 | 7 [2, 12] | 52 [31, 74] | 378 [108, 648] | 401 [328, 475] |
| h1-balance-hard-v0 | 11 [7, 15] | 79 [51, 107] | 31 [5, 56] | 92 [87, 97] |
| h1-balance-simple-v0 | 9 [6, 12] | 69 [58, 80] | 42 [14, 70] | 205 [166, 244] |
| h1-sit-hard-v0 | 15 [-4, 35] | 235 [154, 315] | 723 [660, 786] | 843 [747, 939] |
| h1-sit-simple-v0 | 19 [9, 28] | 874 [869, 879] | 790 [772, 809] | 931 [924, 938] |
| h1-maze-v0 | 113 [107, 118] | 147 [137, 156] | 244 [106, 383] | 354 [349, 359] |
| h1-crawl-v0 | 248 [176, 319] | 582 [563, 600] | 962 [959, 965] | 973 [972, 974] |
| h1-hurdle-v0 | 4 [3, 5] | 60 [18, 102] | 387 [254, 519] | 344 [245, 443] |
| h1-reach-v0 | 3203 [2824, 3581] | 1409 [998, 1821] | 2654 [1951, 3357] | 8101 [7640, 8563] |
| h1-run-v0 | 4 [2, 6] | 91 [54, 128] | 778 [763, 793] | 820 [815, 824] |
| h1-stand-v0 | 15 [7, 22] | 433 [138, 727] | 798 [779, 817] | 856 [815, 897] |
| h1-walk-v0 | 8 [1, 16] | 33 [22, 45] | 814 [813, 815] | 850 [830, 869] |
| IQM | 0.007 [0.004, 0.012] | 0.134 [0.088, 0.245] | 0.734 [0.510, 0.936] | 0.864 [0.735, 0.976] |
| Median | 0.021 [0.000, 0.047] | 0.284 [0.183, 0.392] | 0.696 [0.536, 0.881] | 0.823 [0.733, 0.920] |
| Mean | 0.022 [0.000, 0.046] | 0.289 [0.207, 0.375] | 0.710 [0.562, 0.858] | 0.825 [0.748, 0.902] |

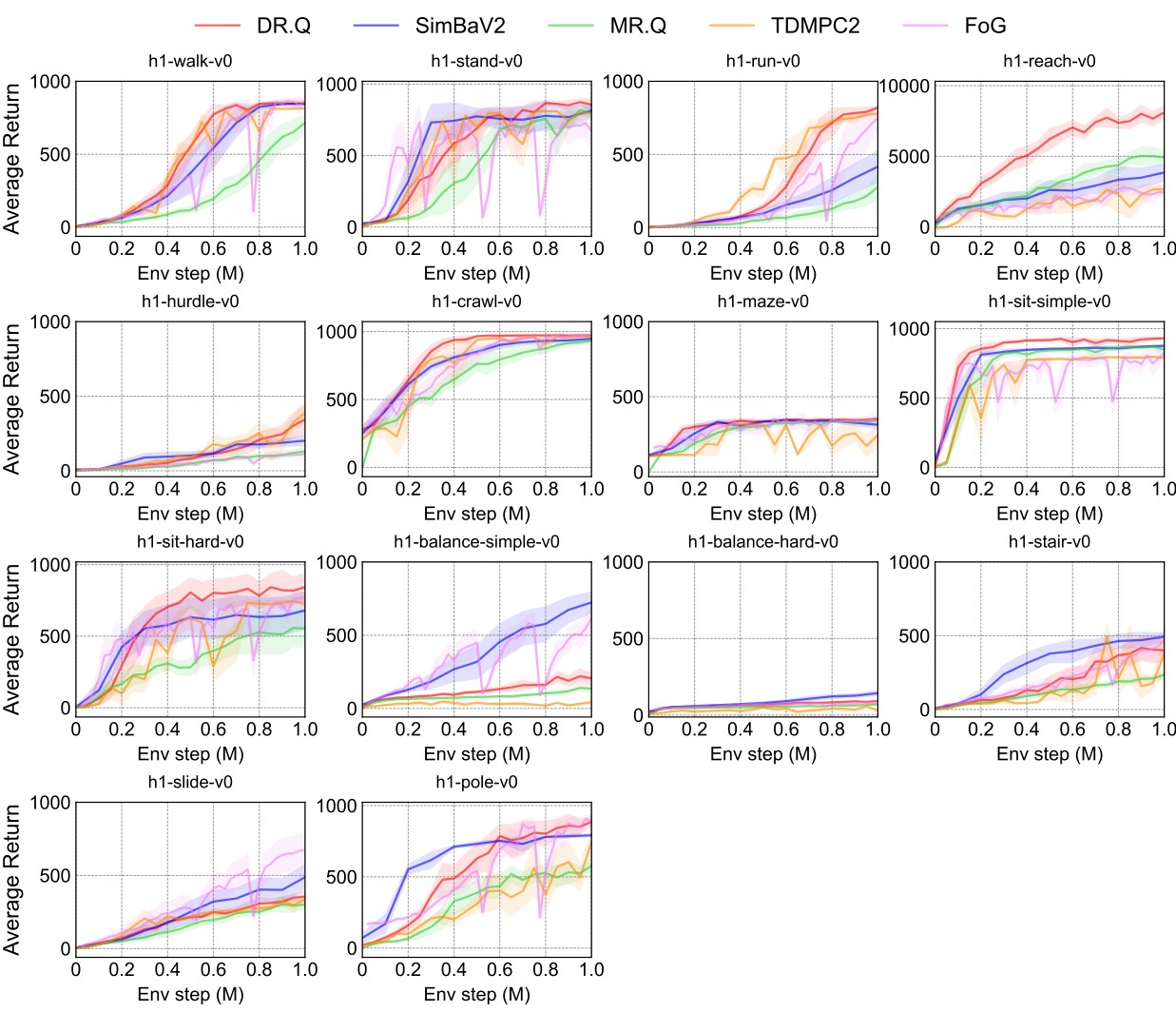

*Figure 8.* **Full learning curves on HumanoidBench (w/o hand) tasks.** We report the average returns (the solid lines) in each task. The light-colored regions denote the 95% bootstrap confidence intervals.

## D.5. HumanoidBench (w/ Hand) Results

*Table 10.* **Full comparison results on HumanoidBench with dexterous hands.** We report the final average return results at 1M environment steps (equivalent to 500K steps due to an action repeat 2). The [bracketed values] denote a 95% bootstrap confidence interval. The aggregate mean, median and interquartile mean (IQM) are computed over the success normalized score.

| Task | DreamerV3 | TDMPC2 | SimBa | SimbaV2 | MR.Q | FoG | DR.Q |
|---|---|---|---|---|---|---|---|
| h1hand-door-v0 | 10 [7, 13] | 134 [23, 246] | 206 [169, 244] | 310 [302, 318] | 293 [280, 305] | 244 [227, 261] | 320 [308, 333] |
| h1hand-slide-v0 | 21 [19, 23] | 79 [68, 90] | 67 [55, 79] | 136 [97, 175] | 146 [131, 161] | 201 [173, 228] | 285 [258, 312] |
| h1hand-stair-v0 | 16 [8, 25] | 43 [35, 51] | 61 [44, 78] | 120 [89, 151] | 127 [104, 150] | 135 [126, 144] | 288 [193, 382] |
| h1hand-bookshelf-simple-v0 | 45 [41, 50] | 97 [59, 134] | 487 [315, 660] | 838 [834, 843] | 691 [599, 783] | 610 [523, 697] | 709 [572, 846] |
| h1hand-bookshelf-hard-v0 | 27 [24, 30] | 34 [19, 50] | 490 [447, 533] | 496 [417, 575] | 332 [240, 425] | 577 [548, 605] | 349 [262, 435] |
| h1hand-sit-simple-v0 | 48 [42, 54] | 607 [268, 947] | 643 [580, 705] | 927 [904, 951] | 653 [568, 737] | 631 [528, 735] | 942 [926, 958] |
| h1hand-sit-hard-v0 | 15 [11, 20] | 139 [86, 193] | 649 [500, 797] | 724 [609, 838] | 487 [353, 621] | 179 [128, 229] | 891 [841, 941] |
| h1hand-basketball-v0 | 13 [12, 13] | 47 [21, 73] | 54 [25, 83] | 56 [34, 78] | 53 [34, 72] | 182 [131, 232] | 75 [45, 105] |
| h1hand-pole-v0 | 48 [36, 60] | 99 [87, 111] | 224 [195, 254] | 493 [426, 559] | 237 [202, 273] | 257 [237, 277] | 424 [299, 549] |
| h1hand-crawl-v0 | 256 [244, 268] | 897 [858, 935] | 779 [748, 809] | 640 [549, 732] | 807 [783, 831] | 794 [721, 866] | 526 [477, 574] |
| h1hand-reach-v0 | 864 [578, 1150] | 3610 [2912, 4309] | 3185 [2664, 3707] | 3223 [2703, 3744] | 4101 [3540, 4662] | 2877 [2487, 3267] | 4950 [4280, 5619] |
| h1hand-run-v0 | 6 [4, 8] | 29 [27, 30] | 31 [24, 37] | 30 [22, 38] | 35 [29, 41] | 22 [19, 25] | 129 [77, 181] |
| h1hand-stand-v0 | 41 [38, 44] | 193 [147, 238] | 127 [72, 181] | 103 [81, 126] | 300 [194, 405] | 79 [66, 91] | 491 [344, 638] |
| h1hand-walk-v0 | 19 [12, 27] | 234 [125, 343] | 94 [79, 109] | 64 [52, 76] | 95 [77, 112] | 75 [63, 87] | 512 [371, 652] |
| IQM | 0.019 [0.013, 0.026] | 0.150 [0.091, 0.224] | 0.219 [0.179, 0.267] | 0.298 [0.241, 0.374] | 0.286 [0.245, 0.333] | 0.254 [0.222, 0.285] | 0.452 [0.400, 0.512] |
| Median | 0.021 [0.010, 0.030] | 0.298 [0.147, 0.433] | 0.356 [0.269, 0.413] | 0.420 [0.338, 0.491] | 0.388 [0.313, 0.449] | 0.342 [0.268, 0.395] | 0.529 [0.455, 0.607] |
| Mean | 0.020 [0.011, 0.028] | 0.282 [0.169, 0.413] | 0.345 [0.286, 0.406] | 0.417 [0.356, 0.482] | 0.385 [0.329, 0.443] | 0.336 [0.285, 0.393] | 0.534 [0.473, 0.595] |

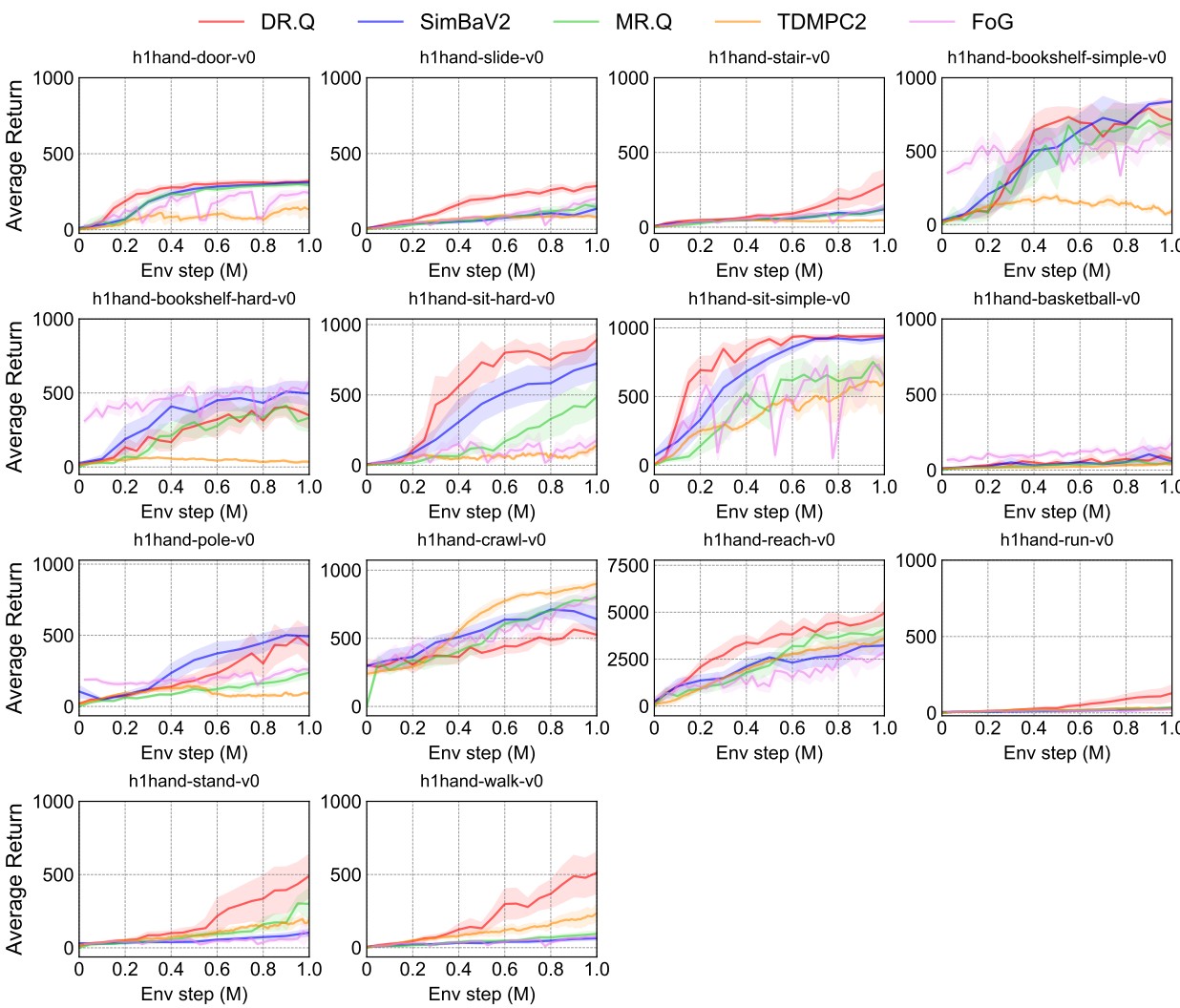

*Figure 9.* **Full learning curves on HumanoidBench (w/ hand) tasks.** We report the average returns (the solid lines) in each task. The light-colored regions denote the 95% bootstrap confidence intervals.

## D.6. Visual DMC Suite Results

Since it is very expensive to run DMC suite tasks with visual inputs, we omit simple tasks where the performance of MR.Q or the baseline methods has already saturated, e.g., `visual-cartpole-balance`, `visual-walker-stand`, etc. We eventually select 12 tasks for experiments, and the overall results across 10 seeds are shown below. DR.Q generally outperforms strong baselines like MR.Q and TDMPC2 in terms of sample efficiency and the final performance, especially on tasks like `visual-dog-stand`, `visual-hopper-stand`, etc.

*Table 11.* **Final performance comparison on DMC suite tasks with visual inputs.** We report the average return results at 1M environment steps (equivalent to 500K steps due to an action repeat 2). The [bracketed values] mean a 95% bootstrap confidence interval. The aggregate mean, median and interquartile mean (IQM) are computed over the success normalized score.

| Task | DrQ-v2 | PPO | TDMPC2 | DreamerV3 | MR.Q | DR.Q |
|---|---|---|---|---|---|---|
| acrobot-swingup | 168 [127, 219] | 2 [1, 4] | 197 [179, 217] | 121 [106, 145] | 287 [254, 316] | 324 [283, 365] |
| dog-run | 10 [9, 12] | 11 [9, 14] | 14 [10, 18] | 9 [6, 14] | 60 [44, 80] | 118 [104, 132] |
| dog-stand | 43 [37, 49] | 51 [48, 56] | 117 [72, 148] | 61 [30, 92] | 216 [201, 232] | 700 [660, 740] |
| dog-trot | 14 [11, 18] | 13 [12, 15] | 20 [14, 25] | 14 [13, 16] | 65 [55, 79] | 113 [98, 128] |
| dog-walk | 22 [18, 29] | 16 [14, 18] | 22 [17, 28] | 11 [11, 12] | 77 [71, 83] | 201 [146, 256] |
| hopper-hop | 224 [170, 278] | 0 [0, 0] | 187 [119, 238] | 205 [125, 287] | 270 [230, 315] | 330 [283, 377] |
| hopper-stand | 917 [903, 931] | 1 [0, 2] | 582 [321, 794] | 888 [875, 900] | 852 [703, 930] | 937 [930, 944] |
| humanoid-run | 1 [1, 1] | 1 [1, 1] | 0 [1, 1] | 1 [1, 1] | 1 [1, 2] | 1 [1,1] |
| quadruped-run | 459 [412, 507] | 118 [98, 139] | 262 [184, 330] | 328 [255, 397] | 498 [476, 522] | 655 [573, 737] |
| quadruped-walk | 750 [699, 796] | 149 [113, 184] | 246 [179, 310] | 316 [260, 379] | 833 [797, 867] | 927 [914, 941] |
| reacher-hard | 705 [580, 831] | 10 [0, 30] | 911 [867, 946] | 338 [227, 461] | 965 [945, 977] | 954 [930, 979] |
| walker-run | 546 [475, 612] | 39 [35, 44] | 665 [566, 719] | 669 [615, 708] | 615 [571, 655] | 746 [713, 778] |
| IQM | 0.241 [0.214, 0.271] | 0.016 [0.013, 0.018] | 0.154 [0.113, 0.224] | 0.168 [0.152, 0.184] | 0.322 [0.239, 0.423] | 0.494 [0.395, 0.604] |
| Median | 0.191 [0.172, 0.211] | 0.013 [0.012, 0.013] | 0.295 [0.198, 0.339] | 0.134 [0.124, 0.198] | 0.398 [0.320, 0.466] | 0.500 [0.427, 0.576] |
| Mean | 0.321 [0.303, 0.340] | 0.034 [0.031, 0.037] | 0.269 [0.214, 0.326] | 0.247 [0.231, 0.262] | 0.395 [0.335, 0.457] | 0.501 [0.439, 0.564] |

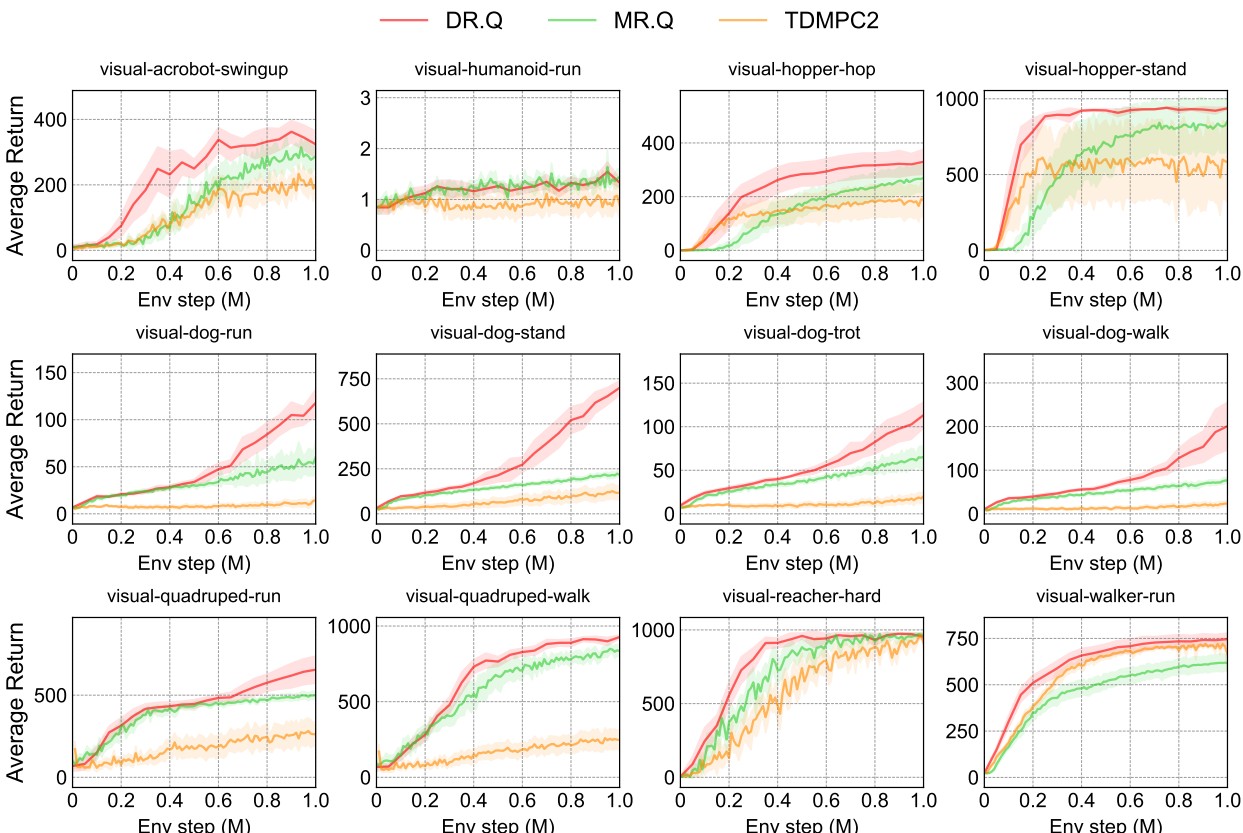

*Figure 10.* **Extended experiments with visual inputs.** We consider selected tasks from DMC suite (with visual inputs) and summarize the results across 10 random seeds. The solid line denotes the average return and the light-colored region is the 95% confidence interval.

# E. Additional Experiments

In this section, we present additional experiments that were omitted from the main paper due to space constraints. For all experiments, we follow the experimental setup described in Section C and run all variants or baselines for 10 seeds and 1M environment steps.

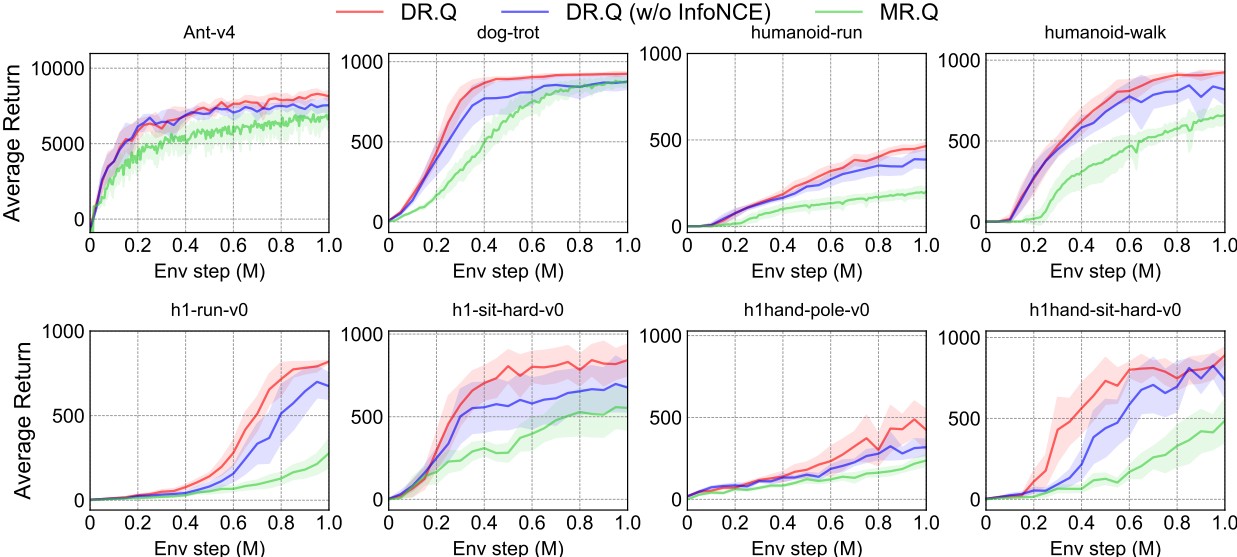

*Figure 11.* **Extended ablation study on InfoNCE loss.** The results are averaged across 10 seeds and the shaded region represents the 95% confidence interval.

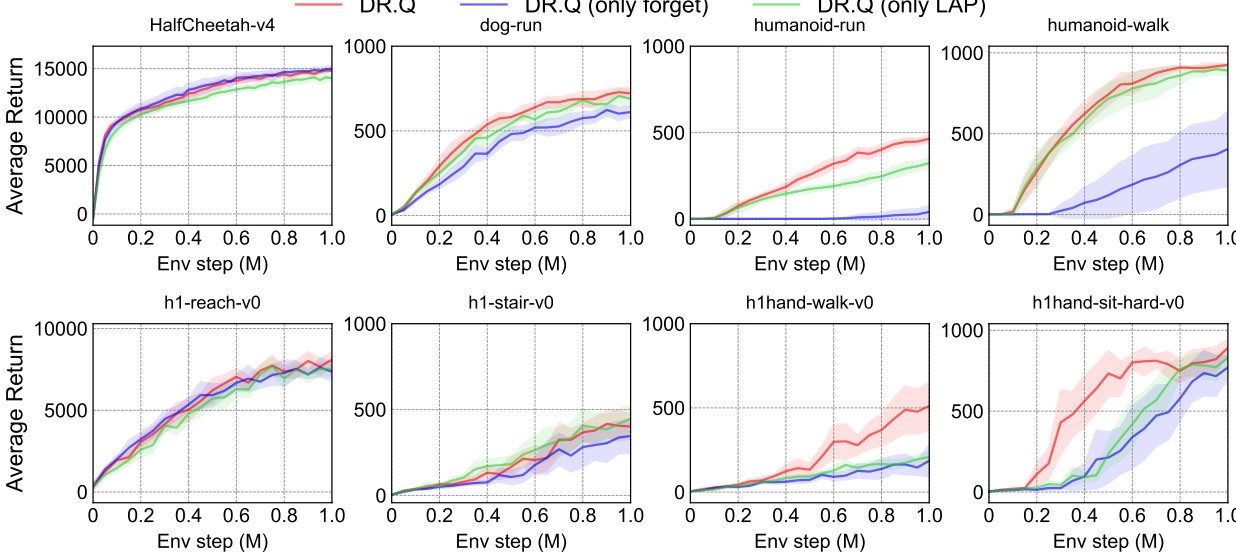

*Figure 12.* **Extended ablation study on sampling strategies.** We report the average return results across 10 seeds and the shaded region is the 95% confidence interval.

## E.1. Extended Ablation Study

We first present the extended ablation study on the InfoNCE loss term and the sampling strategy in Figure 11 and Figure 12. It is easy to find that excluding the InfoNCE loss incurs inferior performance on challenging tasks like h1-sit-hard-v0

and `h1hand-pole-v0`. In almost all evaluated tasks, DR.Q outperforms DR.Q (w/o InfoNCE) in terms of sample efficiency and final performance, demonstrating the necessity and importance of the InfoNCE loss term.

Furthermore, we clearly observe in Figure 12 that excluding either LAP or the forget mechanism can incur unsatisfactory performance, especially on HumanoidBench tasks with dexterous hands, `h1hand-walk-v0` and `h1hand-sit-hard-v0`. Notably, on tasks like `humanoid-run` and `humanoid-walk`, removing LAP (DR.Q (only LAP)) leads to a severe performance drop, which highlights the importance of LAP.

**The latent dynamics loss term.** In Equation 9, DR.Q introduces the latent dynamics loss term $\mathcal{L}_{\text{dynamics}}(\hat{z}_{s',t}, \tilde{z}_{s',t})$ and the information loss term $\mathcal{L}_{\text{I}}(\hat{z}_{s',t}, \tilde{z}_{s',t})$. MR.Q mainly leverages the latent dynamics loss term, while DR.Q adds the InfoNCE loss term for better and more informative representations. To examine the role of the latent dynamics loss term in DR.Q, we remove it from DR.Q, giving rise to DR.Q (w/o dyn loss). We summarize the comparison results on selected tasks from DMC suite and HumanoidBench in Figure 13. The results show that removing the latent dynamics loss term can sometimes have a minor influence on DR.Q (e.g., in `acrobot-swingup`). Nevertheless, DR.Q (w/o dyn loss) can significantly underperform the vanilla DR.Q on tasks like `h1hand-stair-v0`, `h1hand-pole-v0`, etc. Generally, we observe inferior sample efficiency and the final performance without the latent dynamics loss term. It is hence recommended to include it in the objective of DR.Q.

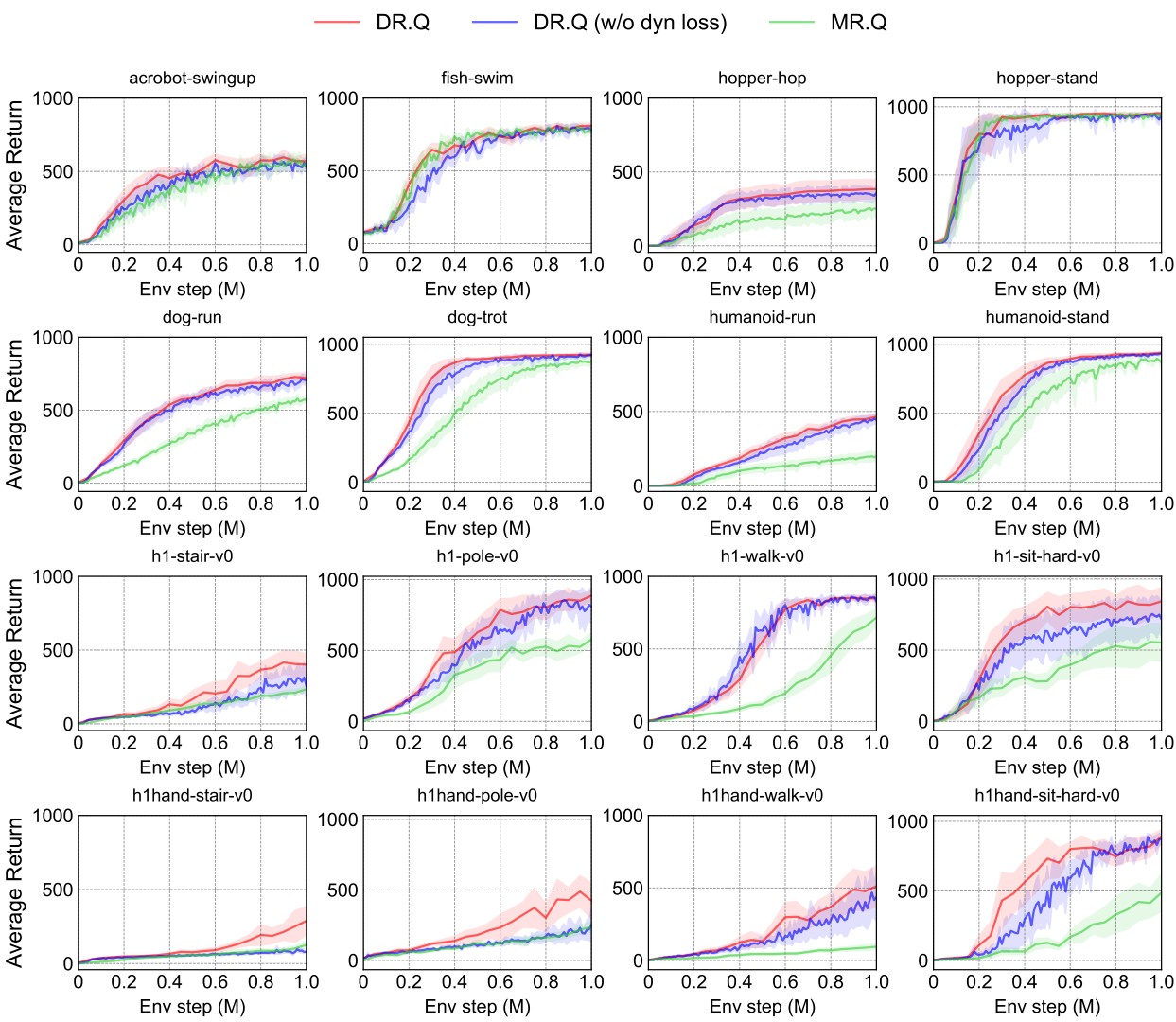

*Figure 13.* **Ablation study on the latent dynamics loss term.** The results are averaged across 10 seeds and the shaded region denotes the 95% confidence interval. Dyn loss denotes the latent dynamics loss.

## E.2. Comparison of DR.Q against MR.Q with Modified Hyperparameters

One may notice that DR.Q adopts slightly different hyperparameters compared to MR.Q, e.g., encoder hidden dimension, weight decay, etc. We adopt a larger encoder learning rate and encoder hidden dimension to scale the network (Nauman et al., 2024; 2025; Lee et al., 2025a;b; Fu et al., 2025; Wang et al., 2025b; Palenicek et al., 2025). Though these values are design choices, we deem it necessary to conduct some empirical experiments to see how these modified hyperparameters can affect the performance. To that end, we set the hyperparameters of MR.Q to be identical as DR.Q (Table 5) and run experiments on selected tasks from DMC suite and HumanoidBench. Empirical results in Figure 14 show that scaling the encoder networks generally helps improve the performance of MR.Q. However, the performance of MR.Q with modified hyperparameters still lags behind that of DR.Q in numerous environments, which clearly shows the effects of the InfoNCE loss and the faded PER sampling strategy.

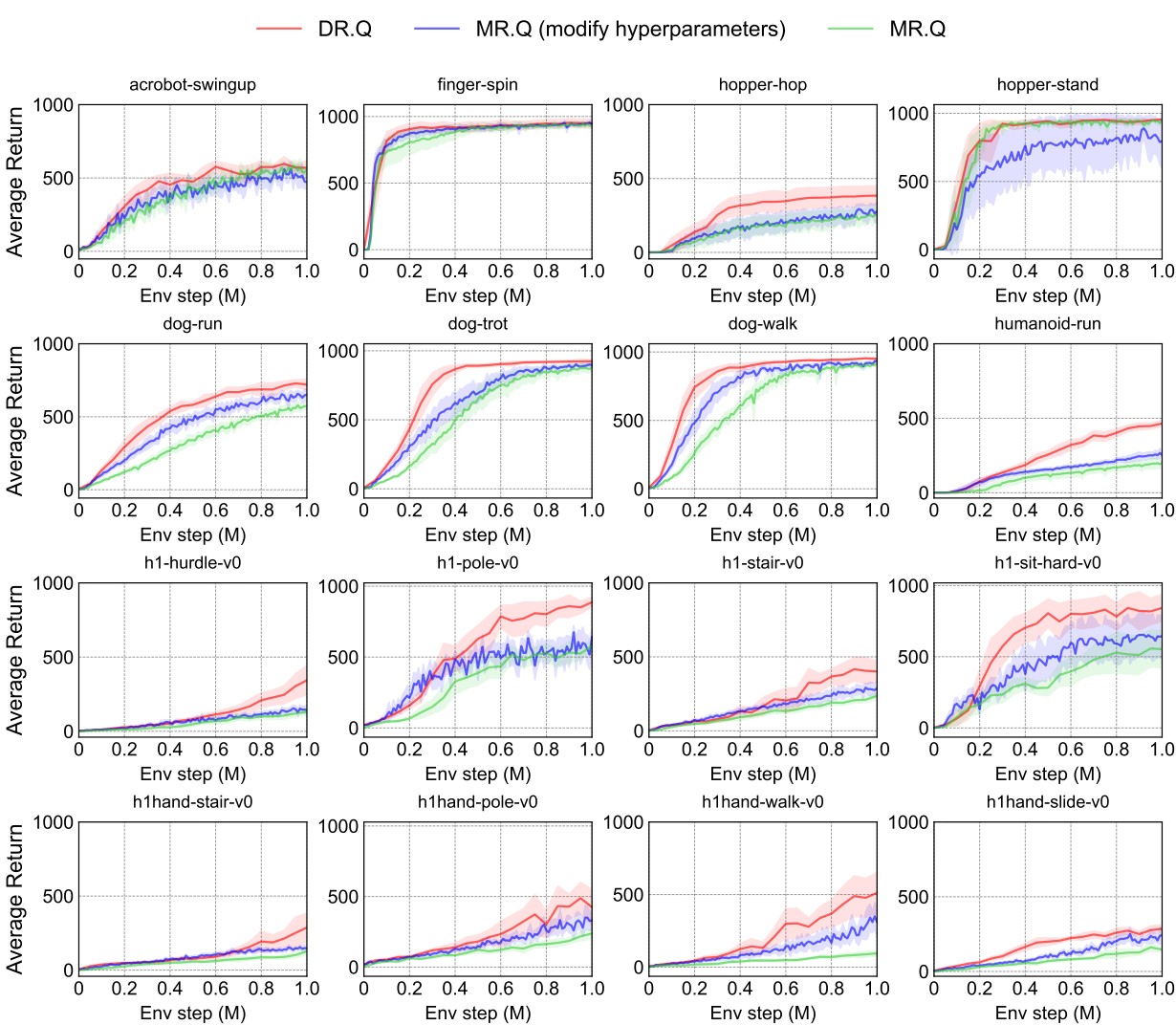

*Figure 14.* **Performance comparison of DR.Q against MR.Q with modified hyperparameters.** The solid line denotes the average return across 10 seeds, and the light-colored region means the 95% confidence interval.

## E.3. On the Mutual Information Loss

**Representation learning with noise.** We demonstrate the necessity and effectiveness of maximizing the mutual information between the state-action representation and the next state representation by additionally considering HumanoidBench (w/o hand) tasks. In this part, we further verify its necessity by expanding the dimensions of the state vectors using noise that is not related to solving the task. To be specific, suppose the original state gives $s$ with shape $d_s$, we construct a 50-dim

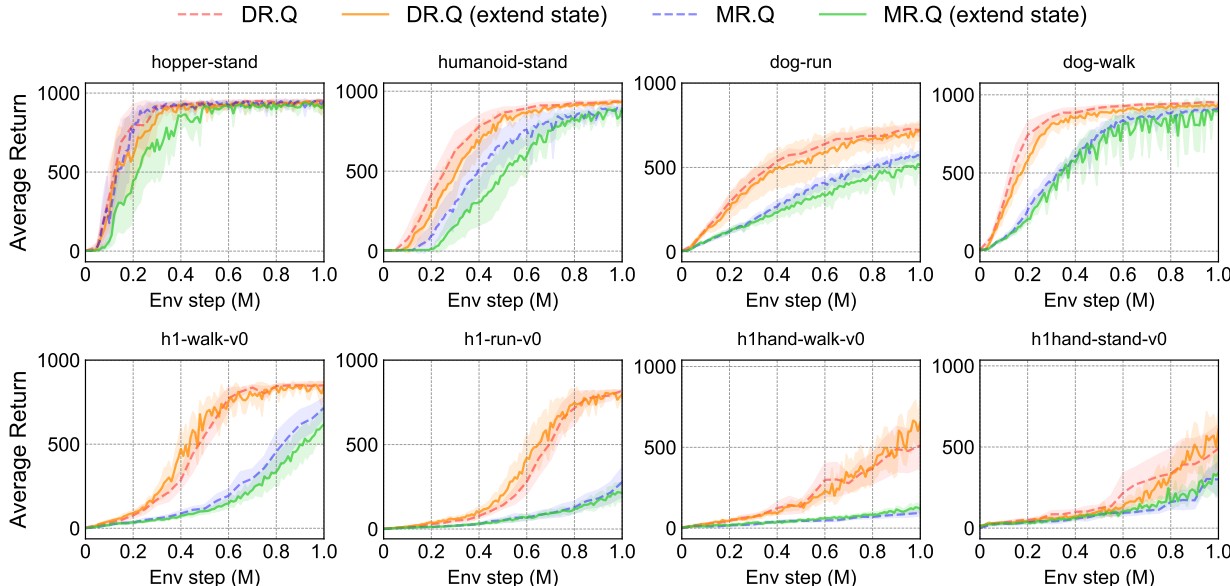

*Figure 15.* **Comparison between DR.Q and MR.Q under extended state inputs.** The dashed lines are the vanilla learning curves of DR.Q and MR.Q using unmodified state inputs and the solid lines denote the learning curves under extended state inputs. The light-colored region captures the 95% confidence intervals. Results are averaged across 10 seeds.

Gaussian noise vector $\Psi = [\psi_1, \ldots, \psi_{50}], \psi_i \sim \mathcal{N}(0, 0.2), i = 1, \ldots, 50$, and concatenate it with the vanilla state vector, resulting in a new state vector with shape $d_s + 50$. In this way, the state input for the agent can contain redundant information. We compare DR.Q against MR.Q on selected challenging tasks from the DMC suite and HumanoidBench and summarize the results in Figure 15. As depicted, extending the state space with noise generally incurs inferior performance and sample efficiency on MR.Q while DR.Q is less affected (e.g., `humanoid-stand`, `hopper-stand`). The performance gap between DR.Q and MR.Q remains significant (e.g., `h1hand-walk-v0`). Note that the performance discrepancy on HumanoidBench tasks may not be large due to the fact that we only inject 50-dimensional noises (dexterous hands introduce 100-dim redundant elements). Still, we observe that the performance of DR.Q does not degrade with additional 50-dimensional noises on HumanoidBench tasks, while MR.Q struggles to achieve strong performance.

**Visualization results.** To further understand how mutual information loss helps improve the learned representations, we conduct a representation analysis using 4 tasks from MuJoCo, DMC suite, and HumanoidBench. To be specific, we visualize the state-action representations $z_{sa}$ via t-SNE (Maaten & Hinton, 2008) after training 200K steps on `HalfCheetah-v4`, `humanoid-walk`, `h1-sit-hard-v0`, `h1hand-stand-v0`. We sample 5000 samples from the replay buffer and feed them to the encoder network for t-SNE visualization in a 2D space. The results are demonstrated in Figure 16. Overall, we observe that MR.Q often incurs separate, discontinuous clusters (e.g., `HalfCheetah-v4`, `humanoid-walk`), while DR.Q exhibits continuous and concentrated clusters, indicating that DR.Q learns better representations than MR.Q. On `h1hand-stand-v0`, MR.Q has two void areas, while DR.Q does not. These observations suggest that DR.Q enables the learning of more structured, more informative and general representations, which contribute to the performance gains.

### E.4. Visualization Results of Sampling Strategies

We now visualize the sampling probability of transitions in the replay buffer to better understand the faded PER method. While MR.Q simply employs LAP for experience replay, where sampling probability is proportional to TD error, DR.Q additionally incorporates a forget mechanism, making its sampling probability proportional to both TD error and the forget weight. We run MR.Q and DR.Q on 8 continuous control tasks for 200K steps, recording the TD errors of the most recent 100K samples. For DR.Q, we use the product of TD error and forget weight as the visualized metric. Figure 17 shows that using LAP alone can lead to large TD errors for old experiences (e.g., in `h1-run-v0`), meaning older samples may be revisited more frequently. In contrast, by integrating the forget mechanism, DR.Q successfully focuses more on recent valuable transitions—those with large TD error but small time indices. Consequently, no old experience attains a higher sampling probability than newer ones. These findings align well with the illustration in Figure 2.

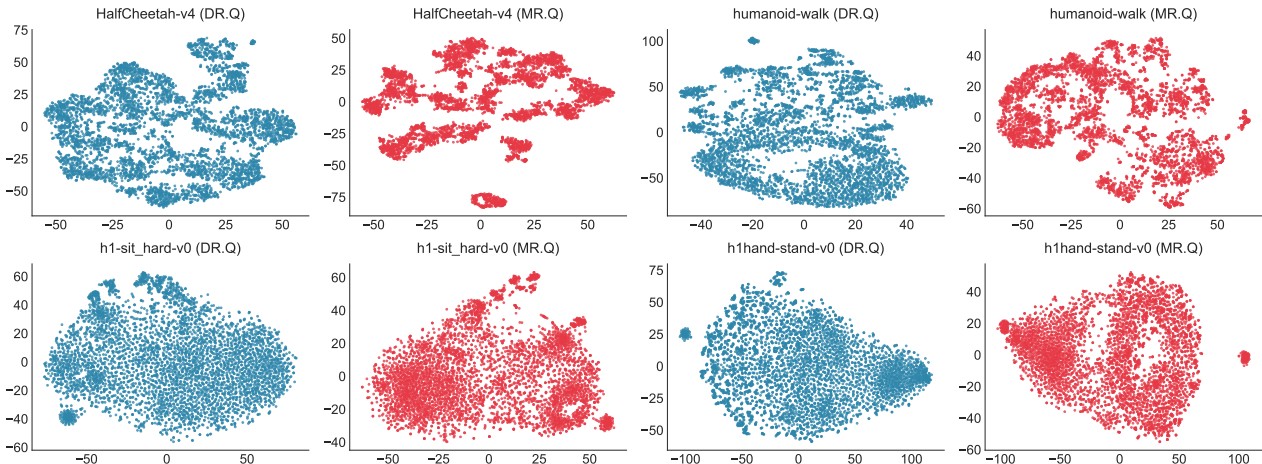

*Figure 16.* **T-SNE visualization results of representations.** The red dots denote the representation produced by MR.Q while the blue dots are representations output by DR.Q.

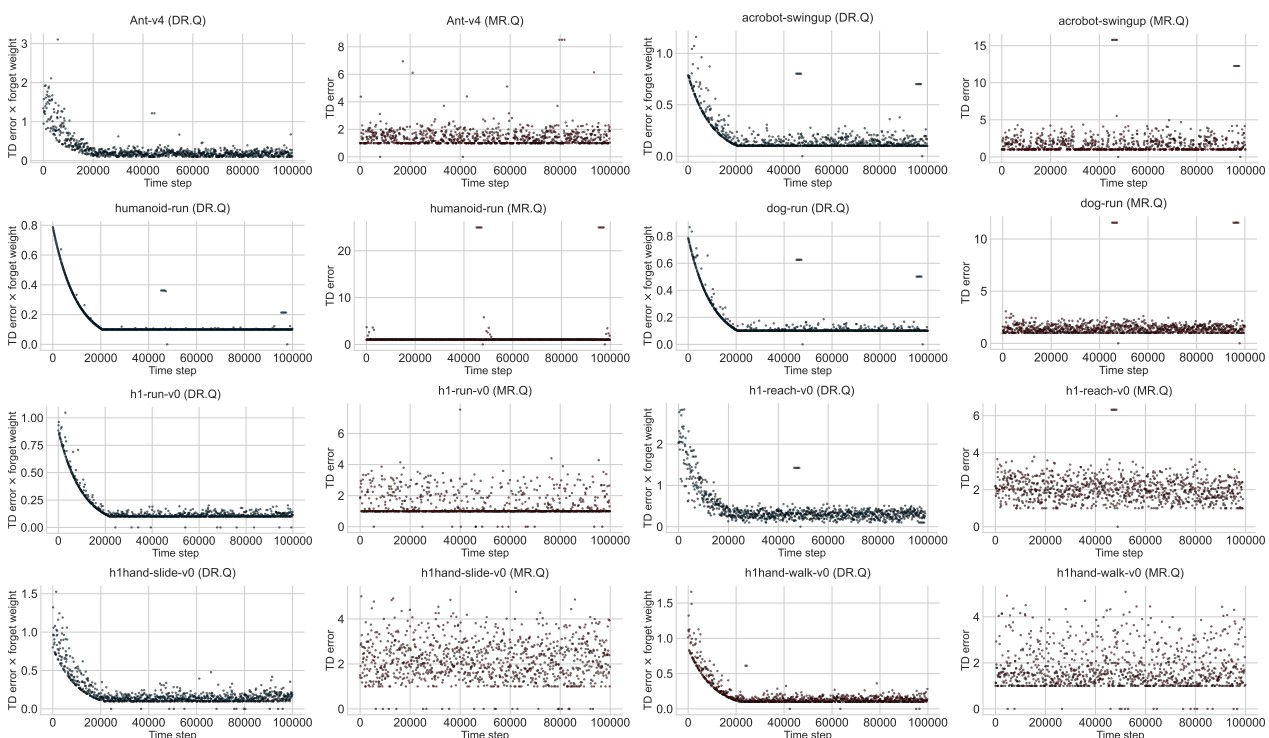

*Figure 17.* **Visualization results of sampling strategies.** The red dots mean the sample point of MR.Q and the blue dots are the samples from DR.Q. We use the first 100K transitions in the replay buffer, with time step 0 being the newest transition.

