# OpenReview forum: "Debiased Model-based Representations for Sample-efficient Continuous Control"
_ICML.cc/2026/Conference — ICML 2026 regular_

### Official Review · Reviewer_x3AP · 2026-03-04

**Soundness:** 4
**Presentation:** 2
**Significance:** 3
**Originality:** 2
**Overall Recommendation:** 5
**Confidence:** 4

**Summary:**

The paper proposes model-based representations in line with prior work such as TDMPC and MR.Q. The core contributions are: extending MR.Q with an auxiliary loss that maximizes the mutual information between representations of the current state-action pair and the next state, and combining PER with a forgetting mechanism, which the authors call Faded Prioritized Experience Replay (Faded PER).

**Compliance With Llm Reviewing Policy:**

Affirmed.

**Ethical Review Concerns:**

The authors addressed my concerns and questions. They improved the motivation for the method and the reasoning on why high MI is desirable. I increased the score from 4 to 5

**Final Justification:**

The authors addressed my concerns and questions. They improved the motivation for the method and the reasoning on why high MI is desirable. I increased the score from 4 to 5.

**Key Questions For Authors:**

Q1: Can you provide a justification for why high MI leads to better downstream policy learning besides the "falsely aligned" argument?

Q2: How do you connect and differentiate your method to relevant methods in contrastive representation learning and self-supervised representation learning?

Q3: Since DR.Q works without the InfoNCE auxiliary loss, representation collapse does not seem to be a critical failure mode here. Do you still believe it is the main driver of the performance gains, and if not, what other factors do you attribute them to?

**Limitations:**

yes

**Strengths And Weaknesses:**

**Strengths**

- Strong results and careful benchmarking across 73 tasks from three continuous control benchmarks. Evaluation is rigorous, reporting median, mean, and IQM.
- Careful ablations cover all important aspects of the method (the dynamics loss term, hyperparameter choices relative to MR.Q, DR.Q without InfoNCE, PER alone, forget-only, and no LAP)

**Weaknesses**

- While the argument that minimizing MSE does not necessarily increase MI is sound, the paper does not sufficiently justify why high MI between representations is desirable in the first place. The theoretical motivation for this remains underexplained.
- Missing connections to the representation learning literature. Several closely related lines of work are not discussed or differentiated:
	- Contrastive representation learning in RL:  CURL (Srinivas et al.) and ATC (Stooke et al.) both employ InfoNCE losses for representation learning in RL. TACO (Zheng et al.) is also directly relevant.
	- Self-supervised representation learning and collapse: Methods in line of BYOL (Grill et al.)  are prone to representation collapse due to the absence of a reconstructive supervisory signal, as used for instance in Dreamer. TDMPC addresses this through regularization via auxiliary reward, value, and action prediction. BLAST (Paster et al.) instead uses BatchNorm and EMA targets with stop-gradient. Joint-embedding predictive architectures such as V-JEPA (Bardes et al.) and IWM (Garrido et al.) are similarly relevant here. The proposed InfoNCE regularization fits naturally into this line of work. Representation collapse does not appear to be a catastrophic failure mode in this setting, as DR.Q functions without the auxiliary loss, but the motivation for the proposed objective nonetheless echoes the same concerns, and a more explicit discussion would strengthen the paper.

---

> ### Author Rebuttal · Authors · 2026-03-30
>
> We thank the reviewer for the insightful feedback. Please check our clarifications to the concerns below
>
> **Concern 1: why high MI is desired**
>
> High MI between state-action representations and next-state representations is desirable because it preserves the predictive and causal information required for downstream value estimation and policy optimization in RL. Meanwhile, maximizing $I(Z\_{sa};Z\_{s^\prime})$ ensures the latent embedding retains sufficient discriminative information about how actions induce state transitions, preventing the loss of dynamical structure that is critical for modeling the environment’s transition dynamics and avoiding degenerate representations that break the predictive link between actions and outcomes. Consequently, high MI between representations helps produce more informative and dynamics-aware representations that can possibly benefit value function training and policy training. We will expand the discussion on this in the final version of this work
>
> **Concern 2: connections to the representation learning literature**
>
> We appreciate the suggested references and will include a dedicated discussion in the camera-ready version. All these methods aim to produce better and higher-quality representations. Below, we carefully differentiate DR.Q from relevant methods:
>
> - contrastive representation learning methods: CURL conducts two data augmentation methods on the input frames to construct the key and query pairs. The InfoNCE loss here functions to extract high-quality features from raw pixels. ATC adopts the InfoNCE loss between the sampled observation $o_t$ and its near-future observation $o\_{t+k}$. It involves components like data augmentation and a momentum encoder. Both CURL and ATC do not include any dynamics information when learning representations. Instead, DR.Q learns task-relevant and dynamics-aware representations. TACO aims to maximize MI between representations of current states paired with action sequences and representations of the corresponding future states. It involves the environmental dynamics information in training, but it requires action sequences. Simply put, TACO utilizes $(s\_t,[a\_t,a\_{t+1},\ldots,a\_{t+k-1}],s\_{t+k})$ for the InfoNCE loss, while DR.Q uses $(s\_t,a\_{t+1},s\_{t+1})$ and additionally enforces the latent dynamics consistency
> - self-supervised representation learning (SSL) methods: methods in line of BYOL rely on two neural networks that interact and learn from each other, e.g., BYOL adopts two data augmentation tricks on the same image to construct two representations and make them closer. Similarly, TDMPC employs the latent consistency loss that enforces the state-action representation to be close to the target next state representation (these representations are produced by two networks). BLAST uses the recurrent state space model (RSSM) along with BatchNorm and EMA targets with stop-gradient (DR.Q does not use RSSM and other tricks). Methods like V-JEPA and IWM employ joint-embedding predictive architectures to predict the missing part of an input, while DR.Q does not. Different from prior works in SSL methods, DR.Q introduces the InfoNCE loss besides the latent dynamics consistency loss that encourages the state-action representation to encode more relevant and informative information about the next state representation. Another vital difference is that DR.Q additionally uses the faded PER strategy to avoid overfitting early samples with poor qualities
>
> **Concern 3: is MI the main driver of the performance gains**
>
> We agree that DR.Q can still exhibit strong performance without the InfoNCE loss since the overall loss (Eq 9) includes the regularization term like reward loss, and we also have value loss akin to TDMPC. The representation collapse issue should not be severe here. Meanwhile, DR.Q can be viewed as being built upon MR.Q, which already exhibits strong performance across numerous benchmarks. This can also explain that DR.Q can work without the InfoNCE loss
>
> We clarify that while representation collapse is not the primary failure mode (due to the auxiliary reconstruction/consistency losses), representation bias is the critical bottleneck, where the representations may contain redundant information. This hinders further performance improvement and we propose to mitigate this with MI loss. Our ablation studies in Fig 4 and 11 show that excluding the InfoNCE loss incurs a performance drop on all evaluated tasks (sometimes the performance decrease can be severe), indicating that MI loss is definitely one of the critical parts of the performance gains
>
> We further note that the strong performance of DR.Q is a joint result of both the mutual information loss (reducing representation redundancy) and the faded PER sampling strategy (mitigating sampling bias). These two factors are the main drivers of performance gains
>
> Hopefully, these can resolve the concerns. If there is still something unclear, please let us know!

---

> > ### Author Rebuttal · Reviewer_x3AP · 2026-04-04
> >
> > My concerns and questions have been largely addressed.
> >
> > However, the motivation for high MI in the final paper is still unclear (Q1). The theoretical argument currently shows that MSE does not maximize MI (Theorem 4.1) and that MI reduces conditional entropy (Lemma 4.2), so let's add a MI loss. But there is no connection to policy learning. The paper concludes that this "can hopefully benefit actor-critic learning", which is not a justification.
> >
> > I would recommend two directions: empirically, citing works where MI-based objectives outperform MSE in RL; and analytically, looking into work formalizing the connection between representation quality and policy learning. DeepMDP (Gelada et al., 2019) could be a starting point. Without this analytical connection, Theorem 4.1 and Lemma 4.2 seem a bit odd. If the paper chooses a theoretical explanation, the chain of reasoning should go all the way to policy learning, not stop halfway.

---

> > > ### Author Response · Authors · 2026-04-06
> > >
> > > We are glad that the reviewer's concerns are largely addressed. Thank you for the insightful follow-up and the suggestion to bridge the gap between MI and policy learning. We provide the following further clarification on why high MI is desired.
> > >
> > > ## Theoretical connection
> > >
> > > We thank the reviewer for recommending DeepMDP [1], which indeed offers a good start. The core insight of DeepMDP, and that of related works like MR.Q [2], is that value error is upper-bounded by the transition and reward modeling errors in the latent space. Within the framework of MR.Q (Theorem 2) and DeepMDP (Theorem 1 and Theorem 2), a more precise (less uncertain) latent dynamics directly tightens the bound on the value error.
> > >
> > > As shown in our Lemma 4.2, maximizing $I(Z\_{sa}; Z\_{s'})$ strictly reduces the conditional entropy $H(Z\_{s'}|Z\_{sa})$, which implies that the latent dynamics model becomes more deterministic and discriminative when predicting the next state representation. Since we are simultaneously minimizing MSE, the accuracy of the estimated dynamics can be improved, and therefore we can better control the value error upper bound derived in DeepMDP and MR.Q. This provides the analytical link the reviewer sought: high MI $\to$ lower latent entropy $\to$ precise latent dynamics modeling $\to$ tighter value error bound $\to$ superior policy performance, which justifies our MI objective. We will extend discussions in Theorem 4.1 and Lemma 4.2 to explicitly mention the value error bound (referencing the theoretical analysis in DeepMDP and MR.Q) if the reviewer deems it necessary.
> > >
> > > We further respectfully clarify that our reasoning chain in the original submission goes: pitfalls of MSE when learning model-based representations (false alignment, Line 135-149) $\to$ high MI (to debias representation bias) $\to$ the benefits of high MI (Lemma 4.2). Please note that we use conservative terms like "can hopefully benefit actor-critic learning" to avoid any possible overclaims and maintain scientific rigor.
> > >
> > > ## Empirical connection
> > >
> > > Better representations can lead to better outcomes (e.g., policy learning). This has been justified in numerous prior works (e.g., TD7, MR.Q, TDMPC, DrQ-v2). We deem that high MI incurs better representations. This is empirically justified in previous works on contrastive representation learning RL methods (e.g., CURL, ATC, TACO), where they use the MI objective rather than MSE. Since we are dealing with state-action representations and next state representations (previous works like CURL deal with key-query pairs), it is natural that we follow this line of work and seek high MI between them. Note that high MI is also preferred in the context of offline RL [3], multi-agent RL [4], and meta-RL [5], as well as in numerous other areas apart from RL. We will follow the reviewer's suggestion and cite those papers in the final version.
> > >
> > > Learning dynamics-aware representations provides rich, task-relevant information for value estimate and can benefit downstream policy learning. By introducing MI loss, DR.Q filters out task-irrelevant information (redundancy), resulting in more informative latents. This debiasing facilitates faster reward propagation and better performance. Our visualizations in Fig 16 confirm that DR.Q learns more structured and continuous latent clusters compared to the fragmented clusters in MR.Q. This indicates that the MI loss ensures the representation is not just *close* in Euclidean distance (MSE), but *aligned* in information content, directly facilitating more stable TD-updates for the actor-critic.
> > >
> > > All of the comments from the reviewer will be incorporated into the final version of our paper. We thank the reviewer for the efforts and time in making our paper better!
> > >
> > > ## References
> > >
> > > [1] Deepmdp: Learning continuous latent space models for representation learning
> > >
> > > [2] Towards general-purpose model-free reinforcement learning
> > >
> > > [3] Mutual information regularized offline reinforcement learning
> > >
> > > [4] A maximum mutual information framework for multi-agent reinforcement learning
> > >
> > > [5] Domino: Decomposed mutual information optimization for generalized context in meta-reinforcement learning

---

### Official Review · Reviewer_YDdV · 2026-03-12

**Soundness:** 3
**Presentation:** 3
**Significance:** 3
**Originality:** 3
**Overall Recommendation:** 5
**Confidence:** 3

**Summary:**

A central concept studied by the manuscript is the bias in existing model-based representation learning for off-policy continuous control, which arises from insufficient information capture in representations and overfitting to early experiences in the replay buffer. The authors focus on the concept of debiasing these representations to boost the sample efficiency and performance of RL agents. They propose DR.Q, an algorithm that adds an InfoNCE loss to maximize mutual information between state-action and next-state representations, paired with a faded prioritized experience replay strategy to alleviate primacy bias. Evaluated on 73 continuous control tasks across 3 benchmarks, DR.Q matches or outperforms state-of-the-art baselines with a single set of hyperparameters.

**Compliance With Llm Reviewing Policy:**

Affirmed.

**Final Justification:**

concern addressed.

**Key Questions For Authors:**

1) How is the InfoNCE loss weight λm=0.1 determined, and what is its sensitivity across low-dimensional and high-dimensional tasks?
2) DR.Q achieves significant gains on high-dimensional HumanoidBench tasks with dexterous hands, is there direct quantitative evidence that the mutual information loss is the core driver of this improvement?
3) DR.Q shows inferior performance on some challenging vision-based control tasks, and its effectiveness has not been validated in discrete action settings. Can the two core debiasing designs of DR.Q (the mutual information representation loss and faded PER) be directly transferred to reinforcement learning tasks with discrete action spaces? If not, what critical algorithm adaptations and modifications are required to accommodate the inherent properties of discrete action spaces?
4) Can the code/data be open-sourced after the paper is accepted?

**Limitations:**

see questions

**Strengths And Weaknesses:**

Strengths:
1) The method is sound. Extracts high-quality representations via the debiased model, which improves the performance of the final policy；
2) The experiments are comprehensive and technically solid；
3) Theoretically proves that minimizing representation deviation does not guarantee higher mutual information, providing a rigorous foundation for the core design.
4) The two debiasing modules are lightweight and easy to implement, with comprehensive ablation studies verifying their effectiveness

Weaknesses:
1) Lacks quantitative analysis of how the mutual information loss reduces representation redundancy in high-dimensional state spaces.
2) Fixed hyperparameters lead to inferior performance on specific tasks (e.g., Hopper-v4), with no sensitivity analysis of key hyperparameters like the InfoNCE weight and PER decay rate.

---

> ### Author Rebuttal · Authors · 2026-03-30
>
> We thank the reviewer for commenting that our work is sound and solid. Please check our clarifications to the concerns below
>
> **Concern 1: mutual information and redundancy reduction**
>
> MI loss encourages the model to capture task-relevant factors and de-emphasize redundant information, resulting in more informative representations. In Appendix Fig 16, we provide t-SNE visualizations comparing DR.Q and MR.Q. MR.Q representations often form fragmented, discontinuous clusters, while DR.Q exhibits more concentrated and continuous structures. This improved latent structure directly reflects the reduction of redundancy
>
> **Concern 2: hyperparameter sensitivity**
>
> Our design philosophy favors a general-purpose configuration (one set of hyperparameters). InfoNCE loss weight $\lambda\_m=0.1$ is determined by conducting initial experiments on selected tasks with various $\lambda\_m$. We choose a value that is generally acceptable to the community and can incur overall good performance. We agree that hyperparameter sensitivity matters and conduct experiments. Results below show that the optimal $\lambda\_m$ varies across tasks, while $\lambda\_m=0.1$ exhibits the best overall performance
>
> |Env|DR.Q|DR.Q ($\lambda\_m=0.01$)|DR.Q ($\lambda\_m=0.5$)|DR.Q ($\lambda\_m=1$)|
> |----|:---:|:---:|:---:|:---:|
> |acrobot-swingup|**569**±81|541±74|515±77|485±72|
> |dog-run|721±59|664±72|**740**±39|726±52|
> |dog-trot|**925**±18|923±15|913±40|906±53|
> |dog-stand|972±15|**974**±11|969±12|962±14|
> |dog-walk|950±13|939±16|**951**±8|947±13|
> |humanoid-run|**465**±33|454±43|463±38|438±63|
> |humanoid-stand|**938**±10|932±11|933±12|931±15|
> |humanoid-walk|**925**±11|911±24|910±41|921±14|
> |h1-run|**820**±7|803±43|810±13|809±28|
> |h1-sit\_simple|**931**±11|913±27|911±18|889±85|
> |h1-sit\_hard|**843**±154|718±217|731±209|830±149|
> |h1-stair|**401**±118|348±87|340±52|399±118|
> |h1-stand|856±67|827±67|868±18|**877**±48|
> |h1-walk|**850**±31|813±80|843±16|841±39|
> |h1hand-run|129±84|143±102|**157**±121|131±53 |
> |h1hand-sit\_simple|942±26|934±13|915±56|**943**±11|
> |h1hand-sit\_hard|891±81|**895**±50|736±239|804±210|
> |h1hand-stair|**288**±152|221±40|213±80|192±36|
> |h1hand-stand|491±236|511±225|445±208|**542**±201|
> |h1hand-walk|**512**±226|472±207|500±218|509±247|
>
> As for PER decay rate, DR.Q decays the transition by 0.9999 by default. Results below show that decaying with a small value causes over-forgetting and inferior performance, while a larger decay rate can incur a large performance drop on some tasks
>
> |Env|DR.Q|DR.Q (decay by 0.999)|DR.Q (decay by 0.99999)|
> |----|:---:|:---:|:---:|
> |acrobot-swingup|569±81|546±66|**571**±69|
> |dog-run|721±59|704±46|**738**±56|
> |dog-trot|**925**±18|908±30|914±30|
> |dog-stand|**972**±15|956±34|967±18|
> |dog-walk|**950**±13|941±13|935±23|
> |humanoid-run|**465**±33|385±67|457±61|
> |humanoid-stand|**938**±10|932±13|930±20|
> |humanoid-walk|**925**±11|917±13|920±7|
> |h1-run|**820**±7|810±8|816±13|
> |h1-sit\_simple|**931**±11|924±18|928±16|
> |h1-sit\_hard|**843**±154|802±157|807±171|
> |h1-stair|**401**±118|347±45|294±49|
> |h1-stand|**856**±67|840±37|847±44|
> |h1-walk|850±31|750±175|**854**±25|
> |h1hand-run|**129**±84|42±16|55±13|
> |h1hand-sit\_simple|942±26|947±13|**950**±21|
> |h1hand-sit\_hard|**891**±81|781±251|852±113|
> |h1hand-stair|**288**±152|182±82|216±129|
> |h1hand-stand|**491**±236|350±208|345±182|
> |h1hand-walk|**512**±226|369±151|307±104|
>
> **Concern 3: driver of the performance improvement**
>
> T-SNE visualization results in Appendix Fig 16 show that MR.Q’s latent space contains void areas on h1hand-stand-v0, while DR.Q produces more structured representations. Empirically, ablation studies in Fig 4 and 11 show that excluding MI loss incurs performance drop on numerous tasks, indicating the necessity of MI loss
>
> Furthermore, we respectfully clarify that the success of DR.Q on HumanoidBench tasks (w/ hand) stems from both MI loss and faded PER. Ablation studies in Fig 4 and 12 show that removing the forget mechanism also incurs large performance degradation
>
> **Concern 4: discrete action tasks**
>
> While our work focuses on continuous control (as per the title), DR.Q can be adapted to discrete action spaces by taking $\arg\max a^\prime$, where $a^\prime$ is defined in Eq. 11, and using Gumbel-Softmax as the final activation rather than Tanh (following MR.Q). We conduct experiments on several Atari tasks (10 seeds). Results below (brackets indicate the 95\%CI) show that DR.Q consistently outperforms MR.Q, sometimes by a large margin, demonstrating its versatility
>
> |Env|MR.Q|DR.Q|
> |----|:---:|:---:|
> |Assault|1296[1254,1343]|**1354**[1276,1431]|
> |Phoenix|5173[5025,5322]|**5222**[5105,5339]|
> |Qbert|3938[3210,4327]| **5304**[4063,6545]|
> |Robotank|13[12,15]|**26**[24,27]|
>
> **Concern 5: open source**
>
> We will open-source the code, log, and model weights on all evaluated tasks. We have included the code and log of DR.Q in the supplementary material
>
> Hopefully, these can resolve the concerns. If there is still something unclear, please let us know!

---

> > ### Author Rebuttal · Reviewer_YDdV · 2026-04-02
> >
> > Thanks for your response. I will raise my scores.

---

> > > ### Author Response · Authors · 2026-04-02
> > >
> > > We are pleased that our response has addressed the reviewer’s concerns. Thank you for raising the score! We sincerely appreciate the reviewer’s positive assessment of our work and the time dedicated to reviewing our paper.

---

### Official Review · Reviewer_yitf · 2026-03-12

**Soundness:** 2
**Presentation:** 3
**Significance:** 2
**Originality:** 2
**Overall Recommendation:** 3
**Confidence:** 4

**Summary:**

This paper proposes improved represenation learning approach for model-based offline RL. The idea is to optimize the mutual information between state-action represenation z(s,a) and the next state representation z(s′), so as to focus on more important factors in the resulting latent represenation compred to traditional MSE-based  optimization objective.

**Compliance With Llm Reviewing Policy:**

Affirmed.

**Key Questions For Authors:**

1. How to adaptatively set the strength of the mutual information based regularization?

2. the represenation itself may be improved in terms of dynamic, however, what we really need is the mutual information between the represenation and the Q* value or optimal policy. How such improvement leads to better performance?

**Limitations:**

yes

**Strengths And Weaknesses:**

strengths

1.	The motivation is clear. The paper points out that using MSE loss in existing representation learning methods may cause performance degradation. It introduces information entropy as a regularization to prevent the optimization process from being misled by irrelevant information in the representation.

2.	The experimental results in Figure 3 demonstrate that the proposed method is effective to some extent in high-dimensional control tasks.

weaknesses

1.	There is a lack of hyperparameter sensitivity analysis in this paper. Table 5 states that the same hyperparameters are used across all environments. However, while the paper introduces information entropy as a loss forces the model to focus on key representations, the degree of redundant information varies across environments. Using the same regularization ratio is thus unreasonable; hyperparameter sensitivity analysis should be added here.

2.	A toy experiment could be supplemented to visualize the distribution changes in the representation space after maximizing mutual information during training, which would help explain the reliability of performance improvements on complex tasks.

3.	The abstract mentions that model-based methods suffer from high complexity, yet introducing InfoNCE also introduces additional overhead. A concrete comparison of computational and memory overhead should be provided.

4.	The proof of Theorem 4.1 has some issues First, in Eq. 18, when k is a non-zero constant, X is completely known, meaning kX has no randomness relative to X; therefore, k does not change the mutual information between them. Based on this specific counter-example, decreasing the MSE loss does not necessarily lead to a decrease in mutual information.

5.	Missing implementation details for Faded PER. The paper should describe specifically how Faded PER is implemented efficiently (e.g., how to update priorities that decay over time without incurring high computational costs).

6.	Unclear motivation for the clipping mechanism. Eq. 10 introduces an additional clipping mechanism to ensure the sampling probability does not fall below \epsilon_{low}, but the motivation is unclear. Does this imply that the forgetting mechanism is otherwise too aggressive, such that clipping is necessary to prevent the over-forgetting of valuable data?

7.	Limited ablation studies. Each component of the proposed method should be experimentally verified for its effectiveness. For example, the role of the stop gradient operator in Eq. 7 and the substitution of MSE loss with Huber loss in Eq. 13 should be ablated.

---

> ### Author Rebuttal · Authors · 2026-03-30
>
> Thank you for the constructive comments. We provide point-to-point clarification to the concerns. If we can resolve some concerns, we hope that the reviewer will be willing to raise the score
>
> **Concern 1: hyperparameter sensitivity**
>
> Our primary goal is to develop a general-purpose algorithm that performs robustly across diverse benchmarks without task-specific tuning. The strong performance of DR.Q across numerous tasks with a fixed $\lambda_m$ validates its robustness. We agree that the degree of redundant information varies across environments, and environment-specific tuning may yield extra performance gains, but our current setup already outperforms strong baselines by a large margin
>
> For sensitivity analysis of the InfoNCE loss weight $\lambda_m$, please refer to Concern 2 of our response to Reviewer YDdV
>
> **Concern 2: visualization of representation changes**
>
> Fig 16 in the appendix offers t-SNE visualizations of DR.Q vs. MR.Q. It clearly demonstrates that the InfoNCE loss leads to more structured latent distributions, validating the effectiveness of mutual information maximization
>
> **Concern 3: computational and memory overhead**
>
> Please see Table 1 of our response to Reviewer z1sA
>
> **Concern 4: on the proof of Theorem 4.1**
>
> We respectfully clarify that $k$ is not a constant, but a variable to optimize (if $k$ is a constant, both MSE and MI are constants, there would be nothing to optimize). Minimizing the MSE encourages $k\to1$. Eq 17 and 18 demonstrate that the MSE and MI will move in the same direction relative to $k$
>
> **Concern 5: faded PER implementation**
>
> We maintain two separate arrays rather than a single priority array that decays over time: priority (update via TD-error) and forget weight (decay over time). When sampling transitions, we multiply these arrays (with clipping) to form a sampling distribution. All computations are performed on GPUs to ensure compute efficiency. We will make it clearer in the camera-ready version
>
> **Concern 6: clipping motivation**
>
> The clipping mechanism is essential to prevent catastrophic forgetting. Without it, the exponential decay is too aggressive; e.g., transitions older than 50k steps would have near-zero forget weight (~0.0067). This decreases the sampling weight of the transition, even if the transition has a large TD error
>
> **Concern 7: limited ablation studies**
>
> As suggested, we conduct additional ablations below
>
> - No Stop-Gradient (NoSG): Removing SG in Eq. 7 destabilizes optimization (similar to why SG is required for critic targets). DR.Q(NoSG) underperforms vanilla DR.Q on many tasks. Note that SG is also used in TD7 and MR.Q when learning model-based representations
> - MSE Loss: Following LAP [1], Huber loss is theoretically preferred to counter bias in prioritized sampling. Our results confirm that replacing Huber with MSE can lead to performance degradation, especially in h1hand tasks
>
> |Env|DR.Q|DR.Q(NoSG)|DR.Q(MSE)|
> |----|:---:|:---:|:---:|
> |dog-run|**721**±59|704±35|695±55|
> |dog-trot|**925**±18|901±55|910±23|
> |dog-walk|**950**±13|945±11|940±15|
> |humanoid-run|**465**±33|441±94|439±40|
> |humanoid-walk|925±11|**930**±11|897±78|
> |humanoid-stand|**938**±10|934±17|922±21|
> |h1-run|**820**±7|801±32|766±119|
> |h1-sit\_simple|**931**±11| 912±37|896±20|
> |h1-sit\_hard| **843**±154|742±194|655±257|
> |h1-stair| **401**±118|358±61|310±49|
> |h1-stand| **856**±67|849±43|804±49|
> |h1hand-run|129±84|**132**±55|45±26|
> |h1hand-sit\_simple|**942**±26|932±38|872±99|
> |h1hand-sit\_hard|**891**±81|810±117|380±337|
> |h1hand-stair|**288**±152|229±58|240±58|
> |h1hand-stand|**491**±236|460±205|296±111|
>
> **Concern 8: adaptively set the strength of MI loss**
>
> The success of our results shows that per-task tuning is not necessary. Nevertheless, one can use a rule-based method to adjust $\lambda\_m$, e.g., set a target value $G$ for the MI loss, denote the MI loss as $l$, then $\lambda\_m\leftarrow 2\times\lambda\_m$ if $l > 1.5G$, and $\lambda\_m\leftarrow 0.5\times \lambda\_m$ if $l< G/1.5$. This idea comes from the PPO paper [2]
>
> **Concern 9: how representation improvement incurs better performance**
>
> Learning dynamics-aware representations provides rich, task-relevant information for value estimate and can possibly benefit downstream policy learning. By introducing MI loss, DR.Q filters out task-irrelevant information (redundancy), resulting in more informative latents. This debiasing facilitates faster reward propagation and better performance. As shown in Fig 16, better representations directly correlate with superior policy performance. Furthermore, the theoretical analysis in the original MR.Q paper shows a connection between the quality of the representation and corresponding dynamics, and the ability to capture the true value function
>
> Hopefully, these can resolve the concerns. If there is still something unclear, please let us know!
>
> [1] An Equivalence between Loss Functions and Non-Uniform Sampling in Experience Replay
>
> [2] Proximal policy optimization algorithms

---

> > ### Author Rebuttal · Reviewer_yitf · 2026-04-02
> >
> > Thank you for the clarifications.

---

> > > ### Author Response · Authors · 2026-04-02
> > >
> > > We thank the reviewer for the time dedicated to reviewing our paper. We are pleased that our response has addressed the reviewer’s concerns. We would appreciate it if the reviewer can reconsider the score

---

### Official Review · Reviewer_z1sA · 2026-03-15

**Soundness:** 3
**Presentation:** 3
**Significance:** 3
**Originality:** 3
**Overall Recommendation:** 5
**Confidence:** 4

**Summary:**

This work aims to improve model-based representation methods by integrating the latent consistency loss with additional mutual information loss for model-based objectives. This formulation allows for enhanced representation to harness the advantages of both model-free and model-based methods. Authors propose Debiased model-based Representations for Q-learning (DR.Q) with contributions as follows:

 * Actively maximize the mutation information between the current state-action pair and the subsequent state by minimizing deviations of learned representations. They augment MSE representation loss with an additional MI loss term. Authors claim minimizing the deviation between the current state-action and next state representation does not necessarily incur higher mutual information, motivating the explicit inclusion of the MI loss.
 * Introduce a faded prioritized experience replay which assigns higher priority to new experiences with large TD errors and lower priority to earlier experiences (combines forget mechanism). Authors motivate this by noting that standard PER can lead to overfitting to suboptimal early experiences.

**Compliance With Llm Reviewing Policy:**

Affirmed.

**Final Justification:**

Concerns are addressed.

**Key Questions For Authors:**

* Please provide the reasoning or justification for the task selection in ablation studies.
* A summary plot similar to Figure 1 is recommended for ablation studies.
* Deeper discussions and reasoning on the task environment with unsuccessful performance such as visual-humanoid-run and Hopper-v4. Visualization with t-SNE might provide more information regarding clustered/smooth transitions leading to good/bad performance.

**Limitations:**

Somehow yes. It is briefly mentioned in Appendix but very shallow. I suggest the authors to move it to the main text with deeper discussions.

**Strengths And Weaknesses:**

### Strengths

* The paper is well written and organized with clear presentation
* Theory is simple, clear, and well support authors claims
* Extensive experiments over a wide range of environments and tasks and well designed ablation studies

### Weakness

* I recommend moving the Limitations section in Appendix F to the main paper and expanding the discussions to provide deeper insights.
* A more detailed discussion of computational complexity and the encoder update frequency would be beneficial.
* Additional analysis of the encoder loss and its relationship to performance (average return) would be a valuable addition (in the appendix).

---

> ### Author Rebuttal · Authors · 2026-03-30
>
> We appreciate the reviewer’s positive assessment of our work. Please find our clarifications to the concerns below
>
> **Concern 1: limitation section**
>
> We agree that the limitations deserve more visibility. We will move the limitation section from Appendix F to the main body (leveraging 1 extra page allowed for camera-ready). We will expand discussions on possible reasons for DR.Q's inferior performance on tasks like Hopper-v4 and visual-humanoid-run, and discuss DR.Q's computational overhead
>
> **Concern 2: computational complexity and encoder update frequency**
>
> DR.Q introduces InfoNCE loss and Faded PER, but remains more efficient than baselines like SimBaV2 or FoG because it maintains a Replay Ratio (UTD) of 1. Faded PER only requires storing a 1D *forget weight* array. InfoNCE computation is minimal since the batch size and representation dimensions are not large. Table 1 shows that DR.Q only adds minor compute overhead over MR.Q
>
> |Method|Time (min)|Memory (GB)|
> |----|:---:|:---:|
> |MR.Q|24.63|2.737|
> |DR.Q|26.27|2.745|
>
> Table 1. Computation time@100K environment steps and memory overhead on HalfCheetah-v4. Results are obtained on a single H20 GPU
>
> Encoder update frequency $T$ indicates that we update encoders for $T$ gradient steps every $T$ environment steps. Following MR.Q, we set $T=250$ by default. Our sensitivity analysis below across 8 tasks shows that $T=250$ provides the best balance; extreme values (too frequent or too sparse) lead to inferior performance
>
> |Env|DR.Q|DR.Q ($T=25$)|DR.Q ($T=2500$)|
> |----|:---:|:---:|:---:|
> |dog-run|**721**±59|673±55|683±94|
> |dog-walk|**950**±13|940±27|947±14|
> |humanoid-run|**465**±33|464±26|342±80|
> |humanoid-walk|**925**±11|923±17|913±12|
> |h1-pole|**887**±55|882±59|870±101|
> |h1-walk|850±31|**856**±21|833±29|
> |h1hand-stair|**288**±153|207±136|195±53|
> |h1hand-sit\_simple|**942**±26|911±40|939±31|
>
> **Concern 3: relationship between encoder loss and performance**
>
> We think there is no direct connection between the numerical value of the encoder loss and the final return. As analyzed in Sec 4.1 (L136-141), a low MSE loss in MR.Q can stem from *false alignment*. The encoder loss of DR.Q is a combination of the reward loss, the latent dynamics consistency loss, and the InfoNCE loss. It is difficult to analyze the relationship between the numerical encoder loss and the agent's performance given these confounding factors. Furthermore, different tasks may incur encoder losses with varied numerical scales, making it hard to interpret their relationship. We can add encoder loss curves and discuss them in the appendix if the reviewer deems it necessary
>
> The success of prior works like MR.Q and TDMPC show that the encoder loss matters for the good performance of the agent. DR.Q demonstrates that augmenting model-based objectives (reward/latent consistency loss) with mutual information further boosts performance. This indicates that representation quality matters. Our visualizations (Fig. 16) show that DR.Q gives better representations compared with MR.Q. We also analyze the influence of the latent consistency loss to the agent's performance by removing it in Fig 13, where we observe inferior performance on many environments. This demonstrates the necessity of the latent consistency loss in the encoder loss
>
> **Concern 4: ablation task selection and summary plot**
>
> To ensure task diversity and avoid biased conclusions, we select 12 representative tasks from 3 domains with varied complexities: 2 MuJoCo, 4 DMC-Hard (2 dog/2 humanoid), and 6 HumanoidBench (3 tasks with/without dexterous hands). While the main text highlights 4 tasks per ablation study, we present additional results in Fig. 11 and 12 (in the appendix). We hence believe the task selection is reasonable. We will be glad to include a summary plot for ablation studies (akin to Fig. 1) in the final version
>
> **Concern 5: analysis of environments with unsuccessful performance**
>
> For the visual-humanoid-run task, all methods fail to achieve meaningful scores because it is challenging. DrQv2 requires ~15M environment steps to achieve meaningful performance on it (Fig 3 in [1]), while we only run DR.Q and baselines for 1M steps. The encoders may not capture good representations for downstream policy and critic learning within such a limited budget
>
> Hopper-v4 task also remains challenging for recent strong methods (e.g., BRO, MR.Q, TDMPC2, FoG). As DR.Q is built upon MR.Q, the current mutual information and Faded PER components are insufficient to overcome the specific optimization bottlenecks of this environment without further architectural changes or extensive hyperparameter tuning (e.g., InfoNCE loss weight). This can be the cost of the fixed hyperparameter setup, which we deem tolerable, as DR.Q exhibits strong performance on most tasks
>
> [1] Mastering visual continuous control: Improved data-augmented reinforcement learning
>
> Hopefully, these can resolve the concerns. If there is still something unclear, please let us know!

---

> > ### Author Rebuttal · Reviewer_z1sA · 2026-04-03
> >
> > Thank you for the response.

---

> > > ### Author Response · Authors · 2026-04-03
> > >
> > > We are glad that the reviewer's concerns are addressed. Thanks for your time and efforts in making our paper better.

---

### Decision · Program_Chairs · 2026-04-30

**Decision:**

Accept (regular)

**Comment:**

This paper introduces a method to debias model based representations using an InfoNCE loss to maximize mutual information and a faded prioritized experience replay strategy. Reviewers recognized the strong experimental results across numerous continuous control tasks. However, several reviewers raised significant concerns regarding the theoretical motivation for high mutual information, hyperparameter sensitivity, and connections to prior representation learning literature. The authors provided clarifications, including new ablation studies on the InfoNCE loss weight and decay rates, as well as a theoretical grounding linking the mutual information objective to tighter value error bounds via DeepMDP. All four reviewers explicitly confirmed that their concerns were fully resolved. Reviewers YDdV and x3AP explicitly increased their scores after the rebuttal , and while reviewer yitf's numerical score remained at a weak reject, their written confirmation of full resolution indicates a positive shift in consensus. Given the high quality of the reviews, the authors' comprehensive rebuttals, the paper is a strong contribution to the field, resulting in a final recommendation to accept